# Scalars are universal: Equivariant machine learning, structured like classical physics

**Soledad Villar**
Department of Applied Mathematics and Statistics
Johns Hopkins University

**David W. Hogg**
Flatiron Institute
a divison of the Simons Foundation

**Kate Storey-Fisher**
Center for Cosmology and Particle Physics
Department of Physics, New York University

**Weichi Yao**
Department of Technology, Operations, and Statistics
Stern School of Business, New York University

**Ben Blum-Smith**
Center for Data Science
New York University

## Abstract

There has been enormous progress in the last few years in designing neural networks that respect the fundamental symmetries and coordinate freedoms of physical law. Some of these frameworks make use of irreducible representations, some make use of high-order tensor objects, and some apply symmetry-enforcing constraints. Different physical laws obey different combinations of fundamental symmetries, but a large fraction (possibly all) of classical physics is equivariant to translation, rotation, reflection (parity), boost (relativity), and permutations. Here we show that it is simple to parameterize universally approximating polynomial functions that are equivariant under these symmetries, or under the Euclidean, Lorentz, and Poincaré groups, at any dimensionality $d$. The key observation is that nonlinear O($d$)-equivariant (and related-group-equivariant) functions can be universally expressed in terms of a lightweight collection of scalars—scalar products and scalar contractions of the scalar, vector, and tensor inputs. We complement our theory with numerical examples that show that the scalar-based method is simple, efficient, and scalable.

## 1 Introduction

There is a great deal of current interest in building machine-learning methods that respect exact or approximate symmetries, such as translation, rotation, and physical gauge symmetries [18, 49]. Some of this interest is inspired by the great success of convolutional neural networks (CNNs) [51], which are naturally translation equivariant. The implementation of convolutional layers in CNNs has been given significant credit for the success of deep learning, in a domain (natural images) in which the convolutional symmetry is only approximately valid. In many data-analysis problems in astronomy, physics, and chemistry there are *exact* symmetries that must be obeyed by any generalizable law or rule. Since the approximate symmetries introduced by convolutional networks help in the natural-image domain, then we have high hopes for the value of encoding exact symmetries for problems where these symmetries are known to hold exactly.

In detail, the symmetries of physics are legion. Translation symmetry (including conservation of linear momentum), rotation symmetry (including conservation of angular momentum), and time-

35th Conference on Neural Information Processing Systems (NeurIPS 2021).

translation symmetry (including conservation of energy) are the famous symmetries [69]. But there are many more: there is a form of reflection symmetry (charge-parity-time or CPT); there is a symplectic symmetry that permits reinterpretations of positions and momenta; there are Lorentz and Poincaré symmetries (the fundamental symmetries of Galilean relativity and special relativity) that include velocity boosts; there is the generalization of these (general covariance) that applies in curved spacetime; there are symmetries associated with baryon number, lepton number, flavor, and color; and there are dimensional and units symmetries (not usually listed as symmetries, but they are) that restrict what kinds of quantities can be multiplied or added. If it were possible to parameterize a universal or universally approximating function space that is explicitly equivariant under a large set of non-trivial symmetries, that function space would—in some sense—contain within it *every possible law of physics*. It would also provide a basis for good new machine learning methods.

The most expressive approaches to equivariant machine learning make use of irreducible representations of the relevant symmetry groups. Implementing these approaches requires a way to explicitly decompose tensor products of known representations into their irreducible components. This is called the *Clebsch-Gordan problem*. The solution for SO(3) has been implemented, and there is recent exciting progress for other Lie groups [1, 40]. This is an area of current, active research.

Here we give an approach that bypasses the need for a solution to this problem. We find that, for a large class of problems relevant to classical physics, the space of equivariant functions can be constructed from functions only of a complete subset of the scalar products and scalar contractions of the input vectors and tensors. That is, invariant scalars are powerful objects, and point the way to building universally approximating functions that are constrained by exact, non-trivial symmetries.

*Equivariance* is the form of symmetry core to physical law: For an equivariant function, transforming the input results in an output representation transformed in the same way. An *invariant* function, on the other hand, produces the same output for transformed and non-transformed inputs.

*Definition* 1: Given a function $f : \mathcal{X} \to \mathcal{Y}$ and a group $G$ acting on $\mathcal{X}$ and in $\mathcal{Y}$ as $\star$ (possibly the action is defined differently in $\mathcal{X}$ and $\mathcal{Y}$). We say that $f$ is:

$$G\text{-invariant:} \quad f(g \star x) = f(x) \quad \text{for all } x \in \mathcal{X}, \ g \in G \ ; \tag{1}$$

$$G\text{-equivariant:} \quad f(g \star x) = g \star f(x) \quad \text{for all } x \in \mathcal{X}, \ g \in G \ . \tag{2}$$

**Our contributions:** In this work we provide a complete and computationally tractable characterization of all scalar functions $f : (\mathbb{R}^d)^n \to \mathbb{R}$, and of all vector functions $h : (\mathbb{R}^d)^n \to \mathbb{R}^d$ that satisfy all of the symmetries of classical physics. The groups corresponding to these symmetries are given in Table 1; they act according to the rules in Table 2. The characterization we provide is physically principled: It is based on invariant scalars. It is also connected to the symmetries encoded in the Einstein summation rules, a common notation in physics to write expressions compactly but that also allows only equivariant objects to be produced (see Appendix F).

Our characterization is based on simple mathematical observations. The first is the First Fundamental Theorem of Invariant Theory for O($d$): *a function of vector inputs returns an invariant scalar if and only if it can be written as a function only of the invariant scalar products of the input vectors* [93, Section II.A.9]. There is a similar statement for the Lorentz group O($1, d$). The second observation is that *a function of vector inputs returns an equivariant vector if and only if it can be written as a linear combination of invariant scalar functions times the input vectors*. In particular, if $h : (\mathbb{R}^d)^n \to \mathbb{R}^d$ of inputs $v_1, \ldots, v_n$ is O($d$) or O($1, d$)-equivariant, then it can be expressed as:

$$h(v_1, v_2, \ldots, v_n) = \sum_{t=1}^{n} f_t\Big(\langle v_i, v_j \rangle_{i,j=1}^{n}\Big) v_t \ , \tag{3}$$

where $f_t$ can be arbitrary functions, but if $h$ is a polynomial function the $f_t$ can be chosen to be polynomials. In other words, the O($d$) and O($1, d$)-equivariant vector functions are generated as a module over the ring of invariant scalar functions by the projections to each input vector. In this expression, $\langle \cdot, \cdot \rangle$ denotes the invariant scalar product, which can be the usual Euclidean inner product, or the Minkowski inner product defined in terms of a metric $\Lambda$ (see Table 1):

$$\text{Euclidean: } \langle v_i, v_j \rangle = v_i^\top v_j \ , \qquad \text{Minkowski: } \langle v_i, v_j \rangle = v_i^\top \Lambda \, v_j \ . \tag{4}$$

Our mathematical observations lead to a simple characterization for a very general class of equivariant functions, simpler than any based on irreducible representations or the imposition of symmetries

| | |
|---|---|
| Orthogonal | $\mathrm{O}(d) = \{Q \in \mathbb{R}^{d \times d} : Q^\top Q = Q Q^\top = I_d\}$, |
| Rotation | $\mathrm{SO}(d) = \{Q \in \mathbb{R}^{d \times d} : Q^\top Q = Q Q^\top = I_d, \ \det(Q) = 1\}$ |
| Translation | $\mathrm{T}(d) = \{w \in \mathbb{R}^d\}$ |
| Euclidean | $\mathrm{E}(d) = \mathrm{T}(d) \rtimes \mathrm{O}(d)$ |
| Lorentz | $\mathrm{O}(1,d) = \{Q \in \mathbb{R}^{(d+1) \times (d+1)} : Q^\top \Lambda Q = \Lambda, \ \Lambda = \mathrm{diag}([1, -1, \ldots, -1])\}$ |
| Poincaré | $\mathrm{IO}(1,d) = \mathrm{T}(d+1) \rtimes \mathrm{O}(1,d)$ |
| Permutation | $\mathrm{S}_n = \{\sigma : [n] \to [n] \ \text{bijective function}\}$ |

Table 1: **The groups considered in this work.**

| | |
|---|---|
| Orthogonal; Lorentz | $Q \star (v_1, \ldots, v_n) = (Q v_1, \ldots, Q v_n)$ |
| Translation | $w \star (v_1, \ldots, v_n) = (v_1 + w, \ldots, v_k + w, v_{k+1}, \ldots, v_n)$ |
| | (where the first $k$ vectors are position vectors) |
| Euclidean; Poincaré | $(w, Q) \star (v_1, \ldots, v_n) = (Q v_1 + w, \ldots, Q v_k + w, Q v_{k+1}, \ldots, Q v_n)$ |
| Permutation | $\sigma \star (v_1, \ldots, v_n) = (v_{\sigma(1)}, \ldots, v_{\sigma(n)})$ |

Table 2: **The actions of the groups on vectors.** For the Euclidean group, the position vectors are positions of points; for the Poincaré group, the position vectors are positions of *events*.

through constraints (these methods are currently state of the art; see Section 2). This implies that very simple neural networks based on scalar products of input vectors—that enormously generalize those in [80]—can universally approximate invariant and equivariant functions. This justifies the numerical success of the neural network model proposed in [80], and mathematically shows that their method can be extended to a universal equivariant architecture with respect to more general group actions. We note that the formulation in (3) might superficially resemble an attention mechanism [87], but it actually comes from the characterization of invariant functions of the group, in particular the picture for $\mathrm{SO}(d)$-invariant functions is a little different (see Proposition 5).

In Section 2 (and Appendix A) we describe the state of the art for encoding symmetries, the expressive power of graph neural networks, and universal approximation of invariant and equivariant functions. In Section 3 we mathematically characterize the invariant and equivariant functions with respect to the groups in Table 1. In Section 4 we present some examples of physically meaningful equivariant functions, and show how to express them in the parameterization developed in Section 3. In Section 5 and Appendix G we discuss which (of all possible) pairwise inner products one ought to provide, and in Section 6 we discuss some limitations of our approach. We present numerical experiments using our scalar-based approach compared to other methods in Section 7 (see also [96]).

We also note that the symmetries considered in this work are all global symmetries, as they act on all points in the same way. Our characterization thus does not obviously generalize to all gauge symmetries, which are local symmetries that apply changes independently to points at different locations (see [11, 92]). That said, we believe our model could be made general enough to encompass gauge symmetries if we replace the global metric by any position-dependent metric $\Lambda_x$. In this case, spatially separated vectors would need to be propagated to the same point such that they can be input to locally invariant functions, and this can be done with parallel transport. The parallel transport operations would also have to obey our invariance characterization, but we believe this is possible to do; we will explore it in future work.

## 2 Related work

**Group invariant and equivariant neural networks:** Symmetries have been used successfully for learning functions on images, sets, point clouds, and graphs. Neural networks can be designed to parameterize classes of functions satisfying different forms of symmetries, from the classical (approximately) translation-invariant convolutional neural networks [52]—as well as new approaches that enforce additional symmetries (rotation, scale) on these networks [90, 29, 85]—to more recent architectures that define permutation invariant and equivariant functions on point clouds pioneered

by deep sets and pointnets [99, 73, 74], to permutation-equivariant functions on graphs expressed as graph neural networks [33, 81, 47, 23, 30, 15].

For instance, deep sets and pointnets parameterize functions on $(\mathbb{R}^d)^n$ that are invariant or equivariant with respect to the group of permutations $S_n$ acting as in Table 2. Invariant theory shows that all invariant and equivariant functions with respect to such actions can be approximated by easily characterized invariant polynomials [93]. However, the permutation group can act in significantly more complicated ways. For instance, graph neural networks are equivariant with respect to a different action by permutations (conjugation) that is much harder to characterize (see Appendix A).

Since being introduced by [73, 99], deep learning on point clouds has been extremely fruitful, especially in computer vision [36]. Recently, new symmetries and invariances have been incorporated into the design of neural networks on point clouds; especially invariances and equivariances with respect to rigid motions such as translations and rotations. Many architectures have been proposed to satisfy those symmetries such as [85] based on irreducible representations (irreps), [50, 101] based on convolutions, [29] employing spherical harmonics and irreps, [102] using quaternions, and [28] applying a set of constraints to satisfy the symmetries. Most of the implementations of the approaches mentioned above (except for [28]) are limited to 2D or 3D point clouds. We provide an overview of the main approaches below and in Appendix A.

Recently [75] developed an approach to enforcing permutation equivariance in neural network layers using parameter sharing, which can then model other symmetry groups. The weight sharing approach is significantly simpler to implement than the ones described above, and it has been proven to be very successful in practice, with several applications [89, 90], including autonomous driving [38]. Characterizing the space of the functions this approach can express is an interesting open problem.

**Universal approximation via linear invariant layers and irreducible representations:** Universally approximating invariant functions can be obtained by taking universal non-invariant functions and averaging them over the group orbits [97, 68]. However, this approach is not practical for large groups like $S_n$ or infinite groups like O($d$). A classical result in neural networks shows that feed-forward networks with non-polynomial activations can universally approximate continuous functions [54]. This arguably inspired the use of neural networks that are the composition of linear invariant or equivariant layers with compatible non-linear activation functions to create expressive equivariant models [49, 63, 65]. Linear $G$-equivariant functions can be written in terms of the irreducible representations of $G$ (we explain this and refer the reader to related literature in Appendix A). However, the explicit parameterization of the linear maps is only known for a few groups (for instance, SO(3)-equivariant linear maps are parameterized using the Clebsh-Gordan coefficients [29, 85, 8]). Moreover, very recent work [24] shows that the classes of functions defined in terms of neural networks over irreducible representations in [29, 85] are universal. In particular, every continous SO(3)-equivariant function can be approximated uniformly in compacts sets by those neural networks. Despite universality, there is a limitation to this approach: Although decompositions into irreps are broadly studied in mathematics (also as plethysms), the explicit transformation to the irreps is not known or possible for general groups. This is in fact an area of current, active research, where there has been recent exciting progress for other Lie groups [1, 40], but the implementation is still limited.

**Expressive power of (non-universal) neural networks on graphs:** Graph neural networks express functions on graphs that are equivariant with respect to a certain action by permutations. However, the architectures that are used in practice, typically based on message passing [30] or graph convolutions [23, 20], are not universal in general. Implementing a universally approximating graph neural network using the formulation from the previous section would be prohibitively expensive. There is work characterizing the expressive power of message passing neural networks, mainly in terms of the graph isomorphism problem [94, 67, 16, 59, 84, 14], and there is research on the design of graph networks that are expressive enough to perform specific tasks, like solving specific combinatorial optimization problems [6, 13, 70, 44, 95, 43, 9].

**Machine learning for physics with symmetry preservation:** Machine learning has been applied extensively to problems in physics. While many of these problems require certain symmetries—in many cases, exact symmetries—most applied projects to date rely on the data to encode the symmetry and hope that the model learns it. For instance, CNNs are commonly used to classify galaxy images; data augmentation is used to teach the model rotational symmetry [39, 3, 22, 31]. One well-known example is the Kaggle Galaxy Challenge, a classification competition based on the

Galaxy Zoo project [58]; the winning model improved performance by concatenating features from transformed images of each galaxy before further training [21].

There have been recent successes in enforcing physical symmetries in the architecture of the models themselves [98], for instance, in weather and climate modeling [45] and in modeling chaotic dynamical systems such as turbulence [89, 90]. In quantum many-body physics, recent work has shown that the symmetries of quantum energy levels on lattices can be enforced with gauge equivariant and invariant neural networks [17, 10, 88, 60, 78, 61]. There is significant work on imposing permutation symmetry in jet assignment for high-energy particle collider experiments with self-attention networks [27, 53]. In molecular dynamics, rotationally invariant neural networks have been shown to better learn molecular properties [2, 91, 82], and Hamiltonian neural networks have been constructed to better preserve molecular conformations [55]. More broadly, Hamiltonian networks have been shown to improve physical characteristics, such as better conservation of energy, and to better generalize [34, 79, 103], and Lagrangian neural networks can also enforce conservation laws [62, 19].

**Invariant theory as a basis for enforcing symmetry in neural networks:** We are aware of two lines of prior work that develop approaches related to the one taken here: [56, 57] and [35, 37].

In [56] certain tasks in turbulence modeling and materials science are considered, which have built-in O(3) and (in the latter case) octahedral symmetry. Each of these problems involves a specific representation of the given symmetry group, for which an explicit generating set for the invariant algebra (there called an "integrity basis") is known. The authors construct and test models that learn an invariant function, built on these generating sets. In [57], the turbulence example is taken up again, this time with the goal of learning an equivariant 2-tensor.

The idea of using the invariant algebra to enforce physical symmetries is also contemplated in [35]. In this work, the authors are focused on developing the underlying invariant theory in the case of simultaneous Lorentz and permutation invariance. In the followup work [37], this idea is developed into three models, using different generating sets, which are tested against each other.

The present work shares with these works the idea of using invariant-theoretic descriptions of invariant functions to hard-code physical symmetries into a neural network that models a physical system. We work in somewhat greater generality. While [56, 57] focus on specific (small) representations of O(3) and finite extensions, and while [35, 37] are focused on invariant functions and restricted to linearly reductive groups, we consider both invariant and equivariant functions for almost all groups relevant to physics, including groups (Euclidean and Poincaré) that are not reductive, although their invariant theory remains under control.

**Inductive bias benefits of incorporating symmetries:** The value of incorporating exact symmetries in machine learning has been recently established empirically in several applications (see for instance [90]). Mathematical theory has been developed to explain how much one can improve in terms of sample complexity and generalization [66, 7, 26] but many questions remain open.

## 3 Equivariant maps

In this Section we provide a simple characterization of all invariant and equivariant functions with respect to the actions in Table 2 by the groups in Table 1. In what follows, $v_1, v_2, \ldots, v_n$ will be vectors in $\mathbb{R}^d$, $G$ will be a group acting in $\mathbb{R}^d$ and $(\mathbb{R}^d)^n$ as in Table 2, $f : (\mathbb{R}^d)^n \to \mathbb{R}$ will be an invariant function with respect to the action, and $h : (\mathbb{R}^d)^n \to \mathbb{R}^d$ will be an equivariant function with respect to the same action. $V$ will denote a $d \times n$ matrix whose columns are the $v_i$ vectors.

**O($d$) and SO($d$) invariance and equivariance:** The following classical result in invariant theory (e.g., [93, Section II.A.9]) shows that O($d$) invariant functions are functions of the scalars $v_i^\top v_j$.

*Lemma* 1 (First Fundamental Theorem for O($d$)): If $f$ is an O($d$)-invariant scalar function of vector inputs $v_1, \ldots, v_n \in \mathbb{R}^d$, then $f(v_1, v_2, \ldots, v_n)$ can be written as a function of only the scalar products of the $v_i$. That is, there is a function $g(\cdot)$ such that

$$f(v_1, v_2, \ldots, v_n) = g(V^\top V) = g\big((v_i^\top v_j)_{i,j=1}^n\big) . \tag{5}$$

*Proof.* Given $M = V^\top V \in \mathbb{R}^{n \times n}$, we can reconstruct $v_1, \ldots, v_n$ modulo the orthogonal group O($d$) by computing the Cholesky decomposition of $M$ (see for instance [86] p. 174). Therefore, the function $f$ is uniquely determined by the inner product matrix $V^\top V$. $\qquad\square$

The classical theorem also includes the fact that if $f$ is polynomial (in the entries of the $v_j$'s), then $g$ can be taken to be polynomial. In Section 5 we observe that the function $g$ can be determined by a small subset of the scalars $v_i^\top v_j$. There is an analogous statement for SO($d$) (again see [93, Section II.A.9]):

*Lemma* 2 (First Fundamental Theorem for SO($d$))*:* If $f$ is an SO($d$)-invariant scalar function of vector inputs $v_1, v_2, \ldots, v_n \in \mathbb{R}^d$, then $f(v_1, v_2, \ldots, v_n)$ can be written as a function of the scalar products of the $v_i$ and the $d \times d$ subdeterminants of the $d \times n$ matrix $V$.

*Lemma* 3*:* If $h$ is an O($d$)-equivariant vector function of $n$ vector inputs $v_1, v_2, \ldots, v_n$, then $h(v_1, v_2, \ldots, v_n)$ must lie in the subspace spanned by the input vectors $v_1, v_2, \ldots, v_n$.

*Proof.* Let $\{w_1, \ldots, w_r\} \subset \mathbb{R}^d$ be an orthonormal basis of the orthogonal complement to $\text{span}(v_1, \ldots, v_n)$. Then we can write $h(v_1, \ldots, v_n) = \sum_{t=1}^n \alpha_t v_t + \sum_{t=1}^r \beta_t w_t$ for some choice of $\alpha_t, \beta_t$. We claim that the equivariance of $h$ implies $\beta_t = 0$ for $t = 1, \ldots, r$: Consider $\hat{Q} \in O(d)$ such that $\hat{Q}(v) = v$ for all $\text{span}(v_1, \ldots, v_n)$, and $\hat{Q}(w) = -w$ for all $w$ in the orthogonal complement.

Since $(\hat{Q}v_1, \ldots, \hat{Q}v_n) = (v_1, \ldots, v_n)$ we have $h(\hat{Q}v_1, \ldots, \hat{Q}v_n) = \sum_{t=1}^d \alpha_t v_t + \sum_{t=1}^r \beta_t w_t$ while $\hat{Q}(h(v_1, \ldots, v_n)) = \sum_{t=1}^d \alpha_t v_t - \sum_{t=1}^r \beta_t w_t$. Therefore equivariance implies that all $\beta_t = 0$. $\qquad\square$

Note that Lemma 3 doesn't hold for SO($d$): although when the codimension of $\text{span}(v_1, \ldots, v_n)$ is 0 or $\geq 2$ a similar argument works, the situation $\dim_\mathbb{R} \text{span}(v_1, \ldots, v_n) = d - 1$ breaks the proof, and there do exist equivariant vector functions that don't lie in the span when it has codimension 1. For instance, the cross product of two vectors in $\mathbb{R}^3$ is an SO(3)-equivariant function that is not in the span of its inputs. Proposition 5 shows that generalized cross products are the *only* way that SO($d$)-equivariant vector functions can escape the span of the inputs. We further discuss this in Appendix F. Proposition 4 below gives a characterization of all O($d$)-equivariant functions in terms of the scalars. We prove it in Appendix B.

*Proposition* 4*:* If $h$ is an O($d$)-equivariant vector function of $n$ vector inputs $v_1, v_2, \ldots, v_n$, then there are $n$ O($d$)-invariant scalar functions $f_t(\cdot)$ such that

$$h(v_1, v_2, \ldots, v_n) = \sum_{t=1}^n f_t(v_1, v_2, \ldots, v_n) \, v_t \, . \tag{6}$$

Moreover, if $h$ is a polynomial function, the $f_t$ can be chosen to be polynomial.

The equivariant scalar functions form a module over the ring of invariant scalar functions. Proposition 4 states that this module is generated by the projections $(v_1, \ldots, v_n) \mapsto v_j$. Proposition 5 extends this result to SO($d$) in terms of the generalized cross product. The definition of the generalized cross product and the proof of Proposition 5 are in Appendix B.

*Proposition* 5*:* If $h$ is an SO($d$)-equivariant vector function of $n$ vector inputs $v_1, v_2, \ldots, v_n$ then the characterization of Proposition 4 holds, except when $v_1, v_2, \ldots, v_n$ span a $(d-1)$-dimensional space. In that case, there exist O($d$)-invariant scalar functions $f_t(\cdot)$ such that

$$h(v_1, v_2, \ldots, v_n) = \sum_{t=1}^n f_t(v_1, v_2, \ldots, v_n) \, v_t + \sum_{S \in \binom{[n]}{d-1}} f_S(v_1, v_2, \ldots, v_n) \, v_S \, , \tag{7}$$

where $[n] := \{1, \ldots, n\}$, $\binom{[n]}{d-1}$ is the set of all $(d-1)$-subsets of $[n]$, and $v_S$ represents the generalized cross product of vectors $v_j$ with $j \in S$ (taken in ascending order). Moreover, if $h$ is polynomial, the $f_t$ and $f_S$ can be taken to be polynomial.

**E($d$) invariance and equivariance:** When modelling point clouds, we may want to express functions that are translation invariant or equivariant with respect to a subset of the input vectors (for instance, position vectors are translation equivariant). To this end, we consider the group of translations parametrized by $w \in \mathbb{R}^d$ acting on the vectors in $(\mathbb{R}^d)^n$ by translating the position vectors, and leaving everything else unchanged: $w \star (v_1, \ldots, v_k, v_{k+1}, \ldots, v_n) = (v_1 + w, \ldots, v_k + w, v_{k+1}, \ldots, v_n)$. In this Section we characterize all functions that are translation and rotation invariant/equivariant. In the exposition below we assume for simplicity that all vectors are position vectors, but the results generalize trivially to a mix of vectors.

*Lemma* 6: Any translation-invariant function $f : (\mathbb{R}^d)^n \to \mathbb{R}^\ell$ with inputs $v_1, \ldots, v_n$ can be written uniquely as $f(v_1, v_2, \ldots, v_n) = \tilde{f}(v_2 - v_1, \ldots, v_n - v_1)$, where $\tilde{f} : (\mathbb{R}^d)^{n-1} \to \mathbb{R}^\ell$ is an arbitrary function. If $f$ is polynomial, $\tilde{f}$ is polynomial, and vice versa. If $f$ is equivariant for the action of any given subgroup $G \subset GL(n, \mathbb{R})$, then so is $\tilde{f}$, and vice versa.

The proof is given in Appendix C. There is nothing special about subtracting $v_1$; there exist more natural choices to express translation invariant functions. For instance, one classical way in physics to express translation invariance is to take class representatives of the form $(v_1, \ldots, v_n)$ where $\sum v_i = 0$ (for example, subtracting the center of mass). Proposition 7 characterizes the space of O($d$)-equivariant functions that are also translation invariant or translation equivariant. The proof is in Appendix C.

*Proposition* 7: An O($d$)-equivariant function $h : (\mathbb{R}^d)^n \to \mathbb{R}^d$ that is translation-invariant can be written as (6) where the $f_t$ are O($d$) and translation invariant and $\sum_{t=1}^n f_t(v_1, v_2, \ldots, v_n) = 0$. Similarly, if $h$ is translation-equivariant then we can choose $\sum_{t=1}^n f_t(v_1, v_2, \ldots, v_n) = 1$.

**Lorentz symmetry:** The Lorentz group acts on Minkowski spacetime as Lorentz transformations, which keep the metric tensor invariant. Lorentz transformations relate space and time between inertial reference frames, which move at a constant relative velocity; spacetime intervals are invariant across frames. The group is made up of spatial rotations in the three space dimensions and linear velocity "boosts" along each dimension. This set of symmetries—required for special relativity—is not O(4), but rather the non-compact group O(1,3) defined in Table 1.

The characterization of invariant and equivariant functions we obtained for the orthogonal group can be extended to the Lorentz group, obtaining a very similar result, summarized in Proposition 8 and proven in Appendix D.

*Proposition* 8: A continuous function $h : (\mathbb{R}^{d+1})^n \to \mathbb{R}^{d+1}$ is Lorentz-equivariant (with respect to the action in Table 2) if and only if

$$h(v_1, \ldots, v_n) = \sum_{t=1}^n f_t(v_1, \ldots, v_n) \, v_t \tag{8}$$

where $f_t : (\mathbb{R}^{d+1})^n \to \mathbb{R}$ are Lorentz-invariant scalar functions. Moreover, the functions $f_t$ are uniquely determined by the pairwise Minkowski inner product $\langle \cdot, \cdot \rangle$ (4):

$$f_t(v_1, \ldots, v_n) = g_t(\langle v_i, v_j \rangle_{i,j=1}^n) \, . \tag{9}$$

If $h$ is polynomial, the $f_t$ and corresponding $g_t$ can be taken to be polynomial.

**Poincaré symmetry:** The Poincaré group combines translation symmetry with Lorentz symmetry. Together these complete the symmetries of special relativity, forming the full group of spacetime transformations that preserve the Minkowski metric. The generalization from a Lorentz-equivariant formulation to a Poincaré-equivariant formulation is similar to the generalization from O($d$) to E($d$): The *position* vectors take on a special role, in which only *differences* of position can appear as vector inputs to the functions; all other vectors can act unchanged.

Proposition 9 generalizes the results above to the Poincaré group action. As before, we assume that all vectors are position vectors for simplicity. The proof is analogous to the proof of Proposition 7.

*Proposition* 9: A continuous function $h : (\mathbb{R}^{d+1})^n \to \mathbb{R}^{d+1}$ is Poincaré-equivariant (with respect to the action in Table 2) if and only if

$$h(v_1, \ldots, v_n) = \sum_{t=1}^n f_t(v_1, \ldots, v_n) \, v_t \tag{10}$$

where the $f_t : (\mathbb{R}^{d+1})^n \to \mathbb{R}$ are translation and Lorentz-invariant scalar functions, determined as in (9) by the pairwise Minkowski inner products (4), but also satisfying $\sum_{t=1}^n f_t(v_1, v_2, \ldots, v_n) = 1$. Furthermore, if $h$ is polynomial, the $f_t$ and corresponding $g_t$ (as in (9)) can be taken to be polynomial.

**Permutation invariance and equivariance:** Most physics problems are permutation-invariant, in that once you know the masses, sizes, shapes, and so on of the objects in the problem, the physical predictions are invariant to labeling. In particle physics, this invariance is raised to a fundamental symmetry; fundamental particles (like electrons) are identical and exchangeable.

The characterization of permutation-invariant functions with respect to the action in Table 2 is classical [93, pp. 36–39]. Here we prove an extension that describes permutation-invariant vector functions that are also O($d$)-equivariant. The proof is in Appendix E.

*Proposition* 10: Let $h : (\mathbb{R}^d)^n \to \mathbb{R}^d$ be O($d$)-equivariant (or continuous O($1, d-1$)-equivariant) and also permutation-invariant with respect to the action in Table 2. Then $h$ can be written as

$$h(v_1, \ldots, v_n) = \sum_{t=1}^n f(v_t, v_1, \ldots, v_{t-1}, v_{t+1}, \ldots, v_n)\, v_t, \tag{11}$$

where $f : (\mathbb{R}^d)^n \to \mathbb{R}$ is O($d$)-invariant (or O($1, d-1$)-invariant) and permutation-invariant with respect to the last $n-1$ inputs.

Proposition 11 proven in Appendix E extends the characterization above to permutation equivariant functions (see also [35, 48]).

*Proposition* 11: Let $h : (\mathbb{R}^d)^n \to (\mathbb{R}^d)^n$ be O($d$)-equivariant (or continuous O($1, d-1$)-equivariant) and also permutation-equivariant with respect to the action in Table 2. Then $h$ can be written as $h = (h_1, \ldots, h_n)$ where each $h_i : (\mathbb{R}^d)^n \to \mathbb{R}^d$ is O($d$)-equivariant (or continuous O($1, d-1$)-equivariant) and

$$h_i(v_1, \ldots, v_n) = \sum_{t=1}^n f_t^{(i)}(v_1, \ldots, v_n)\, v_t, \tag{12}$$

where all the $f_j^{(i)} : (\mathbb{R}^d)^n \to \mathbb{R}$ are O($d$)-invariant (or O($1, d-1$)-invariant), and for all $i, j = 1, \ldots, n$ and all $\sigma \in S_n$ we have

$$f_{\sigma^{-1}(j)}^{(i)}(v_{\sigma(1)}, \ldots, v_{\sigma(n)}) = f_j^{(\sigma(i))}(v_1, \ldots, \ldots, v_n). \tag{13}$$

## 4 Examples

Here we briefly state two classical physics expressions that obey all the symmetries, and show how to formulate them in terms of invariant scalars.

**Total mechanical energy:** In Newtonian gravity, the total mechanical energy $T$ of $n$ particles with scalar masses $m_i$, vector positions $r_i$, and vector velocities $v_i$ is a scalar[1] function:

$$T = \frac{1}{2} \sum_{i=1}^n m_i \, |v_i|^2 - \frac{1}{2} \sum_{i=1}^n \sum_{\substack{j=1 \\ j \neq i}}^n \frac{G\, m_i\, m_j}{|r_i - r_j|}, \tag{14}$$

where $G$ is Newton's constant (a fundamental constant, and hence scalar). Since $|a| \equiv (a^\top a)^{1/2}$, this expression (14) is manifestly constructed from functions only of scalars $m_i$ and scalar products of vectors. It is also worthy of note that the positions $r_i$ only appear in differences of position.

**Electromagnetic force law:** The total electromagnetic force $F$ acting on a test particle of charge, 3-vector position, and 3-vector velocity $(q, r, v)$ given a set of $n$ other charges $(q_i, r_i, v_i)$ is a vector function [41]:

$$F = \underbrace{\sum_{i=1}^n k\, q\, q_i \, \frac{(r - r_i)}{|r - r_i|^3}}_{\text{electrostatic force}} + \underbrace{\sum_{i=1}^n k\, q\, q_i \, \frac{v \times (v_i \times (r - r_i))}{c^2\, |r - r_i|^3}}_{\text{magnetic force}}, \tag{15}$$

where $k$ is an electromagnetic constant, $c$ is the speed of light, and $a \times b$ represents the cross product that produces a pseudo-vector perpendicular to vectors $a$ and $b$ according to the right-hand rule. This doesn't obviously obey our equivariance requirements, because cross products deliver parity-violating pseudo-vectors; these can't be written in O(3)-equivariant form. However, a cross of a cross of vectors is a vector, so this expression is in fact O(3)-equivariant, as are all forces (because forces must be O(3)-equivariant vectors in order for the theory to be self-consistent).

We can expand the vector triple product using the identity $a \times (b \times c) = (a^\top c)\, b - (a^\top b)\, c$:

$$\begin{aligned} F &= \sum_{i=1}^n k\, q\, q_i \, \frac{(r - r_i)}{|r - r_i|^3} + \sum_{i=1}^n k\, q\, q_i \, \frac{(v^\top (r - r_i))\, v_i - (v^\top v_i)\, (r - r_i)}{c^2\, |r - r_i|^3} \\ &= \sum_{i=1}^n k\, q\, q_i \left(1 - \frac{v^\top v_i}{c^2}\right) \frac{(r - r_i)}{|r - r_i|^3} + \sum_{i=1}^n k\, q\, q_i \, \frac{(v^\top (r - r_i))\, v_i}{c^2\, |r - r_i|^3}, \end{aligned} \tag{16}$$

---

[1] The energy is *not* a scalar in special relativity or general relativity, but it is a scalar in Newtonian physics.

where the quantity $v^\top v_i$ is the scalar product of the velocities. All of the quantities are now straight-forwardly functions of invariant scalar products times the input vectors.

# 5   How many scalars are needed?

Our analysis in Section 3 shows that the invariant and equivariant functions of interest (under actions in Table 2 from groups in Table 1) with input vectors $v_1, \ldots, v_n$, can be expressed in terms of the scalars $\langle v_i, v_j \rangle_{i \geq j=1}^n$ (4). This greatly simplifies the parameterization of such functions, but it significantly increases the number of features if $n \gg d$. In Appendix G we remark that the scalars can be uniquely determined by a small subset of size approximately $(d+1)\,n$. This is related to the rigidity theory of Gram matrices [77] that answers when there exists a unique set of vectors that realize a partial set of distances, and it is closely related to the low rank matrix completion problem [83]. Furthermore, there is a vast literature studying high probability robust reconstruction of all scalars from a random subset via convex optimization techniques [12]. Recently developed optimization techniques on Gram matrices could provide efficient algorithms to learn invariant and equivariant functions [42].

# 6   Limitations and caveats

Although the principal results presented here work for many groups, and work naturally at all spatial dimensions $d$ (unlike methods based on irreps, for example), they do not solve all problems for all use cases of equivariant machine learning. For one, there are myriad groups—and especially discrete groups—that apply to physical and chemical systems where invariant and equivariant functions do not have such a nice characterization. Such is the case for the GNNs discussed in Appendix A.

One example of a situation in which our formulation might not be practical is provided by multipole expansions (for example, those used in the fast multipole method [5] and $n$-body networks [49]). In the fast multipole method, a hierarchical spatial graph is constructed, and high-order tensors are used to aggregate information from lower-level nodes into higher-level nodes. This aggregation is concise and linear when it is performed using high-order tensors; this aggregation is hard (or maybe impossible) when only scalars can be transmitted, without the use of the irreps of the relevant symmetry groups. This is a research direction we are currently exploring.

Another example that suggests that our universal functions might be cumbersome is the current forms in which classical theories—such as electromagnetism and general relativity—are written. For example, in Section 4 we showed that the electromagnetic force law can be written in the form of functions of scalars and scalar products times vectors, but that is *not* how the theory is traditionally written. It is traditionally written in terms of the magnetic field (a pseudo-vector) or the electromagnetic tensor (an order-2 tensor). As another example, general relativity is traditionally written in terms of contractions of an order-4 curvature tensor. That is, although the theories can in principle be written in the forms we suggest, they will in general be much more concise or simple or clear in forms that make use of higher-order or non-equivariant forms.

Finally, although our results apply to many physically relevant groups, they do not encode all of the symmetries of classical physics. For example, one critical symmetry is the dimensional or units symmetry: You cannot add or subtract terms that have different units (positions and forces, for example). This symmetry or consideration has implications for the construction of valid polynomials. It also implies that only dimensionless (unitless) scalar quantities can be the arguments of large classes of nonlinear functions, including exponentials or sigmoids. These additional symmetries must be enforced at present with some additional considerations of network architecture or constraints.

We also note that in this work we have characterized *global* symmetries, which act on each point in the same way. Our characterization does not obviously generalize to gauge symmetries, which are local symmetries that apply changes independently to spatially separated points (see e.g. [11]). That said, we believe our model could encompass general gauge symmetries if we replace the local metric by any position-dependent metric $\Lambda_x$. Recent work has shown that equivariance under gauge symmetries is possible in the realm of convolutional neural networks by defining coordinate-independent kernels [92]. In our case, we would have to propagate spatially separated vectors to the same location in order to pass them to a locally invariant function, requiring the operation of parallel

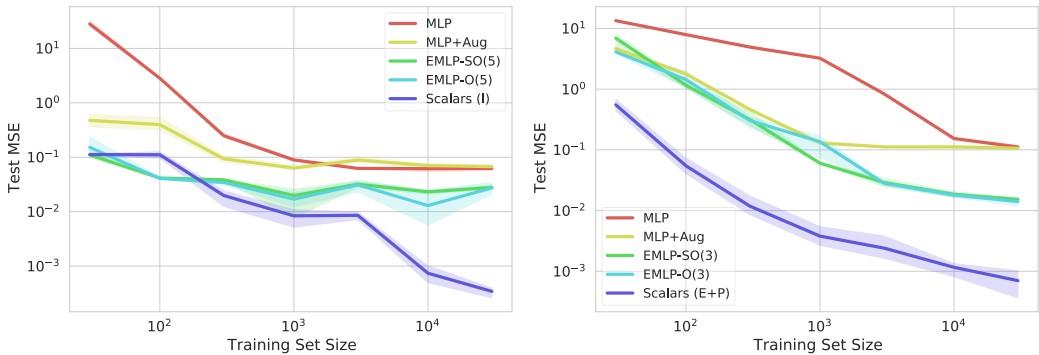

Figure 1: Test error as a function of training set size for (**Left**) the O(5)-invariant task, and (**Right**) the O(3)-equivariant task. Scalars (I) denotes the MLP model using the scalars method for O(5)-invariance construction, and Scalars (E+P) denotes the MLP model using the scalars method for O(3)-equivariance and permutation invariance construction. MLP denotes a standard multilayer perceptron, and MLP+Aug denotes an MLP that has been trained with data augmentation to the given symmetry group. EMLP-G denotes the EMLP models from [28] with different relevant symmetry groups G. For both tasks, the scalar method outperforms all other methods. The shaded regions depict 95% confidence intervals taken over 3 runs.

transport. This operation would then also have to obey our invariance characterization; we believe this is possible and we will detail it in future work.

## 7  Numerical experiments

We demonstrate our approach using scalar-based multi-layer perceptrons (MLP) on two toy learning tasks from [28]: an O(5)-invariant task and an O(3)-equivariant task. Further numerical experiments with these methods applied to dynamical systems appear in [96]. The code is available on GitHub[2], and it reuses much of the functionality provided by EMLP [28].

**O(5)-invariant task:** Given observations of the form $(x_1^i, x_2^i, f(x_1^i, x_2^i))_{i=1}^N$, where $x_1^i, x_2^i \in \mathbb{R}^5$ and $f$ is an O(5)-invariant scalar function, we aim to learn $f$. In this case $f$ is the example from [28]:

$$f(x_1, x_2) = \sin(\|x_1\|) - \frac{\|x_2\|^3}{2} + \frac{x_1^\top x_2}{\|x_1\|\|x_2\|}, \quad x_1, x_2 \in \mathbb{R}^5. \tag{17}$$

We model $f$ using Lemma 1, namely $f(x_1, x_2) = g(x_1^\top x_1, x_1^\top x_2, x_2^\top x_2)$ where $g : (\mathbb{R})^3 \to \mathbb{R}$ could be any function. We learn $g$ by implementing it as an MLP.

**O(3)-equivariant task:** In this task, the data are $n = 5$ point masses and positions $(m_i, x_i)_{i=1}^5$, and the goal is to predict the matrix $\mathcal{I} = \sum_{i=1}^5 m_i(x_i^\top x_i I - x_i x_i^\top)$. To this end we aim to learn

$$h : (\mathbb{R} \times \mathbb{R}^3)^5 \to \mathbb{R}^{3 \times 3}$$
$$(m_i, x_i)_{i=1}^5 \mapsto \mathcal{I}. \tag{18}$$

The function $h$ is O(3)-equivariant in positions, and $S_5$-invariant, namely, for all $Q \in$ O(3) and $\sigma \in S_5$ we have $h((m_{\sigma(i)}, Qx_{\sigma(i)})_{i=1}^5) = Qh((m_i, x_i)_{i=1}^5)$. By Proposition 10 we know there exists functions $f_0$, $f_1$, and $f_2$ such that

$$h((m_i, x_i)_{i=1}^5) = \sum_{i=1}^5 f_0(x_i^\top x_i, m_i, \{x_k^\top x_l, m_k, m_l\}_{k,l \neq i}) \, x_i x_i^\top +$$

$$+ \sum_{i>j=1}^5 f_1(x_i^\top x_j, m_i, m_j, \{x_k^\top x_l, m_k, m_l\}_{k,l \neq i,j}) \, x_i x_j^\top + f_2(\{x_i^\top x_j, m_i, m_j\}_{i,j=1}^5)I.$$

Here the set notation means that the function $f_i$ is permutation-invariant with respect to the inputs in the set. The function $h$ is O(3)-equivariant by construction (see Section 3). We model $f_0$, $f_1$, and $f_2$ with MLPs on deep sets [100].

---

[2] https://github.com/weichiyao/ScalarEMLP

**Acknowledgements:** It is a pleasure to thank Tim Carson (Google), Miles Cranmer (Princeton), Johannes Klicpera (TUM), Risi Kondor (Chicago), Lachlan Lancaster (Princeton), Yuri Tschinkel (NYU), Fedor Bogomolov (NYU), Gregor Kemper (TUM), and Rachel Ward (UT Austin) for valuable comments and discussions. We especially thank Gerald Schwartz (Brandeis) for indicating the line of the polynomial argument to us, and Marc Finzi (NYU) for their help with the EMLP codebase. SV was partially supported by NSF DMS 2044349, the NSF–Simons Research Collaboration on the Mathematical and Scientific Foundations of Deep Learning (MoDL) (NSF DMS 2031985), and the TRIPODS Institute for the Foundations of Graph and Deep Learning at Johns Hopkins University. KSF was supported by the Future Investigators in NASA Earth and Space Science and Technology (FINESST) award number 80NSSC20K1545.

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
