# A Symmetry-enforcing universal neural networks and irreducible representations

In this Section we describe the general principles to parameterize invariant and equivariant universally approximating functions. As mentioned in Section 2, the most common approach is is to write the invariant/equivariant functions as the composition of linear invariant/equivariant layers and non-linear compatible pointwise activation functions. We start by describing the $G$-invariant graph networks from [64], which are invariant with respect to a group $G$ (typically a subgroup of the symmetric group $S_n$, since the work is in the context of graph neural networks) acting on $\mathbb{R}^n$ as $g \star x$ ($g \in G$, $x \in \mathbb{R}^n$). They extend the action $\star$ to tensors $t \in \mathbb{R}^{n^k}$ by acting in each of the $k$ dimensions of the tensor.

Consider a graph $X = (V, E)$ on $n$ nodes. Let $V = (v_1, \ldots, v_n) \in (\mathbb{R}^d)^n$, where $v_i \in \mathbb{R}^d$ are the node features, and assume for simplicity that the edge features are real numbers represented by the matrix $E \in \mathbb{R}^{n \times n}$, then a graph neural network learns equivariant functions $f : X \to (\mathbb{R}^\ell)^n$, where the group of permutations $S_n$ acts in $(\mathbb{R}^d)^n$ and $(\mathbb{R}^\ell)^n$ as in Table 2, and it acts on the graph $X$ as:

$$\Pi \star (V, E) = (\Pi \star V, \Pi \star E) \quad \text{where } \Pi \star E = \Pi E \Pi^\top . \tag{19}$$

The GNN learns an embedding of the form:

$$\mathcal{N}(X) = \theta \circ L_T \circ \ldots \circ L_2 \circ \theta \circ L_1(E) , \tag{20}$$

where $\theta$ is a point-wise non-linearity and $L_i : \mathbb{R}^{n^{\otimes k_i}} \to \mathbb{R}^{n^{\otimes k_{i+1}}}$ is a linear *equivariant* map. When $G$ is the group of permutations, the network (20) can universally approximate all continuous equivariant functions as long as the order of the intermediate tensors can be arbitrarily large [64, 46]. The key insight is that the $k_i$-tensors in the intermediate layers can express all (equivariant) polynomial functions of the input of degree $k_i$. Universality follows from a generalization of the Stone-Weierstrass theorem that states that every continuous equivariant function defined on a compact set can be uniformly approximated as closely as desired by equivariant polynomial functions (for the specific action by permutations) [46, 4]. The disavantage of this approach is that the space of linear equivariant functions $L_i$, even though it is fully characterized, has dimension that grows super-exponentially with the order of the tensors [63].

The approach for general groups is very similar. Group-equivariant neural networks are written as the composition of linear equivariant functions going to higher order tensors, composed with non-linear pointwise activation functions (for general groups there may be restrictions on the activation functions so that equivariance is preserved). Recent work proposes to enforce equivariance of these linear maps by imposing constraints [28]. But in general, the most prevalent approach is to express intermediate layers as linear maps between (group) representations of $G$.

Let $G$ be a group acting on $\mathbb{R}^d$ as $\star$. In representation-theory language, a representation of $G$ is a map $\rho : G \to \mathrm{GL}(V)$ that satisfies $\rho(g_1 g_2) = \rho(g_1)\rho(g_2)$ (where $V$ is a vector space and $\mathrm{GL}(V)$ denotes the automorphisms of $V$, that is, invertible linear maps $V \to V$). The group action $\star$ of $G$ on $\mathbb{R}^d$ is equivalent to the group representation $\rho : G \to \mathrm{GL}(\mathbb{R}^d)$ so that $\rho(g)(v) = g \star v$. We can extend the action $\star$ to the tensor product $(\mathbb{R}^d)^{\otimes k}$ so that the group acts independently in every tensor factor (i.e., in every dimension), namely $\rho_k = \otimes_{r=1}^k \rho : G \to \mathrm{GL}((\mathbb{R}^d)^{\otimes k})$.

The first step is to note that a linear equivariant map $L_i : (\mathbb{R}^d)^{\otimes k_i} \to (\mathbb{R}^d)^{\otimes k_{i+1}}$ corresponds to a map between group representations such that $L_i \circ \rho_{k_i}(g) = \rho_{k_{i+1}}(g) \circ L_i$ for all $g \in G$. Homomorphisms between group representations are easily parametrizable if we decompose the representations in terms of irreps:

$$\rho_{k_i} = \bigoplus_{\ell=1}^{T_{k_i}} \mathcal{T}_\ell . \tag{21}$$

In particular, Schur's Lemma implies that a map between two irreps over $\mathbb{C}$ is either zero or a multiple of the identity.

The equivariant neural-network approach consists in decomposing the group representations in terms of irreps and explicitly parameterizing the maps [49, 85, 29]. In general it is not clear how to decompose an arbitrary group representation into irreps. However in the case where $G = \mathrm{SO}(3)$, the

decomposition of a tensor representation as a sum of irreps is given by the Clebsh-Gordan decomposition:

$$\otimes_{s=1}^{k}\rho_s = \oplus_{\ell=1}^{T}\mathcal{T}_\ell \tag{22}$$

The Clebsh-Gordan decomposition not only gives the decomposition of the RHS of (22) but also it gives the explicit change of coordinates. This decomposition is fundamental for implementing the equivariant 3D point-cloud methods defined in [29, 85, 8]. Moreover, very recent work [24] shows that the classes of functions defined in [29, 85] are universal, meaning that every continous SO(3)-equivariant function can be approximated uniformly in compacts sets by those neural networks. However, there exists a clear limitation to this approach: Even though decompositions into irreps are broadly studied in mathematics (a.k.a. plethysm), the explicit transformation that allows us to write the decomposition of tensor representations into irreps is a hard problem in general. It is called the *Clebsch-Gordan problem*. There is exciting, recent progress on this problem for large classes of groups [1, 40]. The approach taken in the present work sidesteps this problem altogether.

# B  Equivariant functions under rotations and the orthogonal group

**Generalized cross-product:** The generalized cross product of $d-1$ vectors in $\mathbb{R}^d$ is defined to be the Hodge dual of the exterior product of the $d-1$ vectors. Namely, given $v_1, \ldots, v_{d-1}$ the cross product $v_1 \times \ldots \times v_{d-1}$ is the unique vector that satisfies that for all $y \in \mathbb{R}^d$

$$\langle v_1 \times \ldots \times v_{d-1}, y \rangle = \det(v_1, \ldots, v_{d-1}, y), \tag{23}$$

where $\langle \cdot, \cdot \rangle$ denotes the usual inner product in $\mathbb{R}^d$, and $\det(v_1, \ldots, v_d)$ corresponds to the determinant of the matrix with rows $v_1, \ldots, v_d$.

**Proof of Proposition 4:** The purely set-theoretic statement in Proposition 4 and the polynomial statement have independent proofs.

For the set-theoretic statement, let $h : (\mathbb{R}^d)^n \to \mathbb{R}^d$ be an arbitrary O($d$)-equivariant vector function. Lemma 3 shows that for any fixed $n$-tuple of vectors $(v_1, \ldots, v_n)$, there exists an $n$-tuple of real numbers $(a_1, \ldots, a_n)$ such that

$$h(v_1, \ldots, v_n) = \sum_{t=1}^{n} a_t v_t. \tag{24}$$

Pick one representative tuple $(v_1, \ldots, v_n)$ from each O($d$)-orbit on $(\mathbb{R}^d)^n$, and find a corresponding tuple $(a_1, \ldots, a_n)$ satisfying this equation. (Note that the tuple $(a_1, \ldots, a_n)$ thus depends on the orbit; however, this dependence is suppressed in the notation to prevent it from becoming cumbersome.) Define O($d$)-invariant functions $f_t : (\mathbb{R}^d)^n \to \mathbb{R}$ by defining $f_t(v_1, \ldots, v_n) = a_t$ for the $a_t$ corresponding to the chosen representative of $(v_1, \ldots, v_n)$'s orbit. Then

$$h(v_1, \ldots, v_n) = \sum f_t(v_1, \ldots, v_n) v_t \tag{25}$$

at the chosen orbit representatives, and the O($d$)-equivariance of both sides then implies this equation is satisfied everywhere.

For the polynomial statement, now assume $h : (\mathbb{R}^d)^n \to \mathbb{R}^d$ is a *polynomial* O($d$)-equivariant map. Define a new map $\overline{h} : (\mathbb{R}^d)^{n+1} \to \mathbb{R}$ by

$$\overline{h}(v_1, \ldots, v_n, y) = \langle h(v_1, \ldots, v_n), y \rangle, \tag{26}$$

where $\langle \cdot, \cdot \rangle$ is the usual inner product on $\mathbb{R}^d$. Then for any $Q \in$ O($d$), we have

$$\begin{aligned}
\overline{h}(Q\,v_1, \ldots, Q\,v_n, Q\,y) &= \langle Q\,h(v_1, \ldots, v_n), Q\,y \rangle \\
&= \langle h(v_1, \ldots, v_n), y \rangle \\
&= \overline{h}(v_1, \ldots, v_n, y),
\end{aligned}$$

where the first equality is by the definition of $\overline{h}$ and the O($d$)-equivariance of $h$ and the second is the fact that the inner product is preserved by O($d$). In other words, $\overline{h}$ is an O($d$)-invariant scalar function with respect to O($d$)'s natural action on $(\mathbb{R}^d)^{n+1}$.

It follows from the First Fundamental Theorem for O($d$) that $\overline{h}$ is a polynomial in the inner products $\langle v_i, v_j \rangle$, $\langle v_t, y \rangle$, and $\langle y, y \rangle$. On the other hand, by its definition, it is homogeneous of degree 1 in the

coordinates of $y$. It follows that $\langle y, y \rangle$ does not appear this polynomial expression for $\overline{h}$ in terms of the dot products, and furthermore, each term contains some $\langle v_t, y \rangle$ with degree 1, and is otherwise composed of $\langle v_i, v_j \rangle$'s. Grouping the terms according to which $\langle v_t, y \rangle$ each contains, we get

$$\overline{h}(v_1, \ldots, v_n, y) = \sum_{t=1}^{n} g_t(\langle v_i, v_j \rangle_{i,j=1}^{n}) \langle v_t, y \rangle \tag{27}$$

for all $v_1, \ldots, v_n, y$. Defining $f_t(v_1, \ldots, v_n) = g_t(\langle v_i, v_j \rangle_{i,j=1}^{n})$ for each $t$, we find the $f_t$'s are invariant polynomials. Unspooling the definition of $\overline{h}$ on the left side, and using the linearity of the dot product on the right, this becomes

$$\langle h(v_1, \ldots, v_n), y \rangle = \left\langle \left( \sum_{t=1}^{n} f_t(v_1, \ldots, v_n) v_t \right), y \right\rangle. \tag{28}$$

As this equation holds for all $y$, and dot product is a nondegenerate bilinear form, we can conclude that

$$h(v_1, \ldots, v_n) = \sum_{t=1}^{n} f_t(v_1, \ldots, v_n) v_t, \tag{29}$$

with the $f_t$ invariant polynomials, as promised. $\qquad\square$

**Proof of Proposition 5:** The proof of this proposition runs parallel to the proof of Proposition 4. For the set-theoretic part of the proposition, we need a substitute for Lemma 3 that applies to SO($d$). The needed statement is that if $h : (\mathbb{R}^d)^n \to \mathbb{R}^d$ is SO($d$)-equivariant, and the span of $v_1, \ldots, v_n$ has dimension different from $d - 1$, then $h(v_1, \ldots, v_n)$ lies in $\mathrm{span}(v_1, \ldots, v_n)$. We see this as follows:

Let $W = \mathrm{span}(v_1, \ldots, v_n)$, and suppose this has dimension $d - m$. The pointwise stabilizer of $W$ in SO($d$), call it $G_W$, acts in the orthogonal complement $W^\perp$ of $W$. Isomorphically, $G_W$ is SO($m$), and its action on $W^\perp$ is SO($m$)'s canonical action on $\mathbb{R}^m$. If $m \neq 1$, this action is irreducible; in particular, there are no nonzero vectors in $W^\perp$ that are fixed by the whole action, and it follows that there are no vectors in $\mathbb{R}^d$ lying outside of $W$ that are fixed by $G_W$. On the other hand, because $G_W$ fixes each of $v_1, \ldots, v_n$, equivariance of $h$ implies $h(v_1, \ldots, v_n)$ is fixed by $G_W$ as well. It follows that $h(v_1, \ldots, v_n) \in W$.

From this lemma it follows that for any tuple $(v_1, \ldots, v_n)$ with $\dim \mathrm{span}(v_1, \ldots, v_n) \neq d - 1$, there exists a solution in $(a_1, \ldots, a_n) \in \mathbb{R}^n$ to the equation

$$h(v_1, \ldots, v_n) = \sum a_t v_t. \tag{30}$$

On the other hand, when $\dim \mathrm{span}(v_1, \ldots, v_n) = d - 1$, then there must exist $d - 1$ linearly independent vectors $v_{i_1}, \ldots, v_{i_{d-1}}$. In this situation, the generalized cross product $v_{i_1} \times \cdots \times v_{i_{d-1}}$ is nonzero, and linearly independent from $v_{i_1}, \ldots, v_{i_{d-1}}$ (in fact it lies in the orthogonal complement of $\mathrm{span}(v_1, \ldots, v_n)$). Thus it and $v_1, \ldots, v_n$ span $\mathbb{R}^d$. So in all cases, i.e., for any tuple $(v_1, \ldots, v_n)$, there exists a solution in the $a_t$ and $a_S$ to

$$h(v_1, \ldots, v_n) = \sum_{t=1}^{n} a_t v_t + \sum_{S \in \binom{[n]}{d-1}} a_S v_S, \tag{31}$$

where the notation is as in the statement of Proposition 5.

The proof of the set-theoretic statement now proceeds exactly as for the proof of Proposition 4: from each SO($d$)-orbit in $(\mathbb{R}^d)^n$, choose a representative tuple $(v_1, \ldots, v_n)$; pick values of $a_t$ and $a_S$ that satisfy the above for this tuple; use them to define invariant functions $f_t(v_1, \ldots, v_n)$ and $f_S(v_1, \ldots, v_n)$; then the equation

$$h(v_1, \ldots, v_n) = \sum_{t=1}^{n} f_t(v_1, \ldots, v_n) v_t + \sum_{S \in \binom{[n]}{d-1}} f_S(v_1, \ldots, v_n) v_S \tag{32}$$

holds at the chosen representative tuples, and the SO($d$)-equivariance of both sides shows it holds everywhere.

The proof of the polynomial statement is even more directly parallel to the proof of Proposition 4. As in that proof, given an SO($d$)-equivariant polynomial vector function $h : (\mathbb{R}^d)^n \to \mathbb{R}^d$, define a polynomial scalar function

$$\overline{h}(v_1, \ldots, v_n, y) = \langle h(v_1, \ldots, v_n), y \rangle. \tag{33}$$

For the same reason as before, it is SO($d$)-invariant. It follows from the First Fundamental Theorem for SO($d$) that $\overline{h}$ is a polynomial in the dot products $\langle v_i, v_j \rangle$, $\langle v_t, y \rangle$, $\langle y, y \rangle$, and the $d \times d$ subdeterminants $\det(v_{i_1} \ldots v_{i_d})$ and $\det(v_{i_1} \ldots v_{i_{d-1}} \; y)$. Because it is also homogeneous of degree 1 in the coordinates of $y$, $\langle y, y \rangle$ cannot occur, and every term must contain either a $\langle v_t, y \rangle$ or a $\det(v_{i_1} \ldots v_{i_{d-1}} \; y)$ exactly once, and is otherwise a product of $\langle v_i, v_j \rangle$'s and $\det(v_{i_1} \ldots v_{i_d})$'s (which is to say, it is otherwise an SO($d$)-invariant function of the $v_j$'s). Grouping according to which $\langle v_t, y \rangle$ or a $\det(v_{i_1} \ldots v_{i_{d-1}} \; y)$ each term contains, we get

$$\overline{h}(v_1, \ldots, v_n) = \sum_{t=1}^n f_t(v_1, \ldots, v_n) \langle v_t, y \rangle + \sum_{S \in \binom{[n]}{d-1}} f_S(v_1, \ldots, v_n) \det(\tilde{v}_S, y), \tag{34}$$

where the $f_t$'s and $f_S$'s are invariant, and where we have abbreviated $\det(v_{i_1}, \ldots, v_{i_{d-1}}, y)$ as $\det(\tilde{v}_S, y)$ with $S = \{i_1, i_2, \ldots, i_{d-1}\}$. (The tilde is to distinguish it from $v_S = v_{i_1} \times \cdots \times v_{i_{d-1}}$.) Now,

$$\det(\tilde{v}_S, y) = \langle v_S, y \rangle, \tag{35}$$

by definition of the generalized cross product $v_S$, so by applying the linearity of the dot product on the right, and the definition of $\overline{h}$ on the left, we get

$$\langle h(v_1, \ldots, v_n), y \rangle = \Big\langle \sum_{t=1}^n f_t(v_1, \ldots, v_n) v_t + \sum_{S \in \binom{[n]}{d-1}} f_S(v_1, \ldots, v_n) v_S, y \Big\rangle. \tag{36}$$

As with O($d$), we get the equality claimed in Proposition 5 because $y$ is arbitrary and $\langle \cdot, \cdot \rangle$ is a nondegenerate bilinear form. $\qquad \square$

## C  Translation-invariant functions

**Proof of Lemma 6:** Consider the map $\Pi : (\mathbb{R}^d)^n \to (\mathbb{R}^d)^{n-1}$ given by $(v_1, v_2, \ldots, v_n) \mapsto (v_2 - v_1, \ldots, v_n - v_1)$. This map has a section $i : (\mathbb{R}^d)^{n-1} \hookrightarrow (\mathbb{R}^d)^n$ given by $(v_2, \ldots, v_n) \mapsto (0, v_2, \ldots, v_n)$, i.e., $\Pi \circ i$ is the identity. The fibers of $\Pi$ are exactly the orbits of the translation action. Thus a translation-invariant function $f$ on $(\mathbb{R}^d)^n$ descends via $\Pi$ to a well-defined function $\tilde{f}$ on $(\mathbb{R}^d)^{n-1}$, so that

$$f = \tilde{f} \circ \Pi. \tag{37}$$

Then

$$\tilde{f} = \tilde{f} \circ \Pi \circ i = f \circ i. \tag{38}$$

Because $\Pi, i$ are polynomial, these equations show that if either $f$ or $\tilde{f}$ is polynomial then so is the other. Because $i, \Pi$ are both equivariant for the action of $GL(n, \mathbb{R})$, any equivariance property of either $f$ or $\tilde{f}$ with respect to any subgroup $G \subset GL(n, \mathbb{R})$ is passed to the other. $\qquad \square$

**Proof of Proposition 7:** Let $h : (\mathbb{R}^d)^n \to \mathbb{R}^d$ be translation-invariant and O($d$)-equivariant. Using Lemma 6 we can consider an O($d$)-equivariant function $\tilde{h} : (\mathbb{R}^d)^{n-1} \to \mathbb{R}^d$ so that $h(v_1, v_2, \ldots, v_n) = \tilde{h}(v_2 - v_1, \ldots, v_n - v_1)$. Therefore there exists O($d$) invariant functions $\tilde{f}_t$

$$\tilde{h}(v_2 - v_1, \ldots, v_n - v_1) = \sum_{t=2}^n \tilde{f}_t(v_2 - v_1, \ldots, v_n - v_1)(v_t - v_1). \tag{39}$$

This implies $h$ can be written as

$$h(v_1, v_2, \ldots, v_n) = \sum_{t=2}^n f_t(v_1, \ldots, v_n) \, v_t - \Big(\sum_{t=2}^n f_t(v_1, \ldots, v_n)\Big) v_1 \tag{40}$$

where the functions $f_t$ are O($d$)- and translation-invariant, which has the claimed form.

For the case where $h$ is O($d$)- and translation-equivariant we first observe that it suffices to take one representative per orbit, define the function on that representative, and extend it everywhere else by translations:

$$h(v_1, \ldots, v_n) = \tilde{h}(v_2 - v_1, \ldots, v_n - v_1) + v_1, \tag{41}$$

where $\tilde{h}$ is O($d$)-equivariant. Therefore any function $h$ that is O($d$)- and translation-equivariant can be written as

$$h(v_1, \ldots, v_n) = \sum_{t=2}^{n} \tilde{f}_t(v_2 - v_1, \ldots, v_n - v_1)(v_t - v_1) + v_1 \tag{42}$$

$$= \sum_{t=2}^{n} f_t(v_1, v_2 \ldots, v_n) v_t + \left(1 - \sum_{t=2}^{n} f_t(v_1, v_2 \ldots, v_n)\right) v_1. \tag{43}$$

$\square$

## D  Invariant and equivariant functions under the Lorentz group

**Proof of Proposition 8:** The proof follows the pattern of the proof of Proposition 4. The claim for polynomial $h$ follows in exactly the same manner, because there is a First Fundamental Theorem for the Lorentz group precisely analogous to Lemma 1—in fact, both are consequences of the First Fundamental Theorem for O($d$,$\mathbb{C}$) [32, Proposition 5.2.2], because over $\mathbb{C}$, the Lorentz group and the orthogonal group are related by a change of basis (namely, multiplying the space coordinates by $i$).

The claim for arbitrary continuous $h$ has two added subtleties for the Lorentz group. First, unlike for O($d$), the Minkowski inner products of the vectors $v_j$ do not distinguish every pair of distinct O($1,d-1$) orbits from each other: for example the orbit of $(v, \ldots, v)$, with $\langle v, v \rangle = 0$, is indistinguishable from 0 by the inner products, even if $v \neq 0$. So there exist O($1, d-1$)-invariant set functions on $(\mathbb{R}^d)^n$ that are not functions of the Minkowski inner products. However, they do distinguish every pair of *closed* orbits, and every orbit has a closed orbit in its closure (this is a general theorem about the invariant ring of a reductive group, see [72, Theorems 4.6 and 4.7 and their corollaries]; the Lorentz group is reductive because it is a real form of the reductive group O($d, \mathbb{C}$)). Thus every *continuous* invariant function is indeed a function of the Minkowski inner products.

The second subtlety is in proving that a continuous, equivariant vector function $h$ always lies in the span of $v_1, \ldots, v_n$. Our approach in Section 3 to show that the invariant functions under the orthogonal group are restricted to the span of the input vectors is based on the following idea: Given $v_1, \ldots v_n$, if they span $\mathbb{R}^d$, then the result trivially holds. Otherwise let $\{w_1, \ldots w_m\}$ be a basis of span($v_1, \ldots, v_n$) and extend it to a basis of $\mathbb{R}^d$. Then we do a full orthogonalization (via Gram Schmidt) to get orthogonal vectors $u_1, u_2, \ldots, u_d$ and then construct $Q$:

$$u_1 \leftarrow w_1 \tag{44}$$

$$\text{then for each } j \ (2 \leq j \leq d) \text{ in order: } u_j \leftarrow w_j - \sum_{k=1}^{j-1} \frac{w_j^\top u_k}{u_k^\top u_k} u_k \tag{45}$$

$$Q \leftarrow \left[\sum_{j=1}^{m} \frac{1}{u_j^\top u_j} u_j u_j^\top\right] - \left[\sum_{j=m+1}^{d} \frac{1}{u_j^\top u_j} u_j u_j^\top\right]. \tag{46}$$

Then $Q$ is an element of O($d$) that fixes everything in span($v_1, \ldots, v_n$) and does not fix anything outside it. This can be used to show that equivariant functions are restricted to the span of the inputs (see Lemma 3).

This idea can be generalized to the Lorentz group O($1, d$). Let $(t, x_1, \ldots, x_d) \in \mathbb{R}^{d+1}$, the metric is

$$\Lambda = \hat{e}_t \hat{e}_t^\top - \sum_{j=1}^{d} \hat{e}_{x_j} \hat{e}_{x_j}^\top, \tag{47}$$

where $\hat{e}_t$ is a timelike unit vector and the $\hat{e}_{x_j}$ are orthonormal space-like vectors (all orthogonal to $\hat{e}_t$). Given $\{v_1, \ldots v_n\}$ we consider $\{w_1, \ldots w_m\}$ a basis of span($v_1, \ldots, v_n$) and we extend it to $\{w_1, \ldots w_{d+1}\}$ a basis of $\mathbb{R}^{d+1}$ and orthogonalize all the vectors according to the Lorentz generalization of orthogonalization given above to make orthogonal vectors $u_1, u_2, \ldots, u_{d+1}$ and then construct $Q$:

$$u_1 \leftarrow w_1 \tag{48}$$

$$\text{then for each } j \ (2 \leq j \leq d+1) \text{ in order: } u_j \leftarrow w_j - \sum_{k=1}^{j-1} \frac{\langle w_j, u_k \rangle}{\langle u_k, u_k \rangle} u_k \tag{49}$$

$$Q \leftarrow \left[\sum_{j=1}^{m} \frac{1}{\langle u_j, u_j \rangle} u_j u_j^\top \Lambda\right] - \left[\sum_{j=m+1}^{d+1} \frac{1}{\langle u_d, u_d \rangle} u_d u_d^\top \Lambda\right]. \tag{50}$$

Note that (49) and (50) require $\langle u_j, u_j \rangle \neq 0$ for all $j$, however, the Minkowski inner product (4) is not positive definite, in particular there can be *lightlike* vectors $u_j \neq 0$ in which $\langle u_j, u_j \rangle = 0$. In our argument we first will require that if $m = 1$ then $v_1$ is not lightlike. If the procedure (49) generates a lightlike $u_j$ for any $j$ in the range $1 \leq j \leq d+1$ we proceed in the following way:

- If $m > 1$ and $j \leq m$, replace the vectors $w_1, \ldots, w_n$ by a linear combination of them with random coefficients and start again. The fact that the ligthlike vectors are a codimension 1 conic surface ensures that every linear subspace of dimension greater than or equal to 2 has an orthogonal basis where none of the basis elements are lightlike.

- If $j > m$ and $d \geq m + 1$ then the complement of the span of $w_1, \ldots, w_m$ has dimension at least 2 and the same procedure can be applied: one can remove lightlike vectors without changing the subspace by replacing $w_{m+1}, \ldots, w_{d+1}$ with a linear combination of them.

- If $m = d$ the procedure will not produce a lightlike vector as its last vector. This is because an orthogonal basis cannot have a lighlike vector. Each lightlike vector is orthogonal to $d$ linearly independent vectors but one of those is itself. So if $u_{d+1}$ is a lightlike vector $w_{d+1}$ is in the span of $w_1, \ldots w_d$ which contradicts our assumption that $\{w_1, \ldots, w_{d+1}\}$ is linearly independent.

This shows that given $\{v_1, \ldots, v_n\}$ set of vectors spanning a linear space of dimension $m > 1$, there exists a $Q \in O(1, d)$ such that $Q(v) = v$ for all $v \in \mathrm{span}(v_1, \ldots, v_n)$ and $Q(w) \neq w$ for all $w \notin \mathrm{span}(v_1, \ldots, v_n)$, which can be used to show that $h(v_1, \ldots, v_n) \in \mathrm{span}(v_1, \ldots, v_n)$ as in Lemma 3.

If $m = 1$ and $v_1$ is lightlike, the continuity of $h$ and the fact that the set of non-lightlike vectors are dense imply that $h(v_1) \in \mathrm{span}(v_1)$. Without the continuity of $h$ this argument wouldn't work. This is because the set of lightlike vectors is invariant under the action of the Lorentz group. $\qquad\square$

# E  Permutation-invariant and equivariant functions that are also orthogonal or Lorentz-equivariant

**Proof of Proposition 10:** Let $\Pi_{j \to i} = \{\sigma \in S_n : \sigma(j) = i\}$. We start by writing $h$ as in (3). Using the invariance we average over the orbit of $S_n$ we obtain:

$$h(v_1, \ldots, v_n) = \sum_{j=1}^{n} f_j(v_1, \ldots, v_n) v_j \tag{51}$$

$$= \frac{1}{n!} \sum_{j=1}^{n} \sum_{\sigma \in S_n} f_j(v_{\sigma(1)}, \ldots, v_{\sigma(n)}) v_{\sigma(j)} \tag{52}$$

$$= \frac{1}{n!} \sum_{j=1}^{n} \left( \sum_{i=1}^{n} \sum_{\sigma \in \Pi_{j \to i}} f_j(v_{\sigma(1)}, \ldots, v_{\sigma(n)}) v_i \right) \tag{53}$$

$$= \sum_{i=1}^{n} \frac{1}{n!} \left( \sum_{j=1}^{n} \sum_{\sigma \in \Pi_{j \to i}} f_j(v_{\sigma(1)}, \ldots, v_{\sigma(n)}) \right) v_i \tag{54}$$

Let $\tilde{f}_i(v_i, v_1, \ldots, v_{i-1}, v_{i+1}, \ldots, v_n) = \frac{1}{n!} \left( \sum_{j=1}^{n} \sum_{\sigma \in \Pi_{j \to i}} f_j(v_{\sigma(1)}, \ldots, v_{\sigma(n)}) \right)$. For fixed $j$, the set of permutations $\Pi_{j \to i}$ is stable (as a set) under post-composition with any permutation $\tau$ that fixes $i$. As a consequence, each summand $\sum_{\sigma \in \Pi_{j \to i}} f_j(v_{\sigma(1)}, \ldots, v_{\sigma(n)})$ is invariant under the action of any such permutation $\tau$. As a consequence, $\tilde{f}_i$ is invariant with respect to the last $n - 1$ inputs, and $h$ can be expressed as

$$h(v_1, \ldots, v_n) = \sum_{i=1}^{n} \tilde{f}_i(v_i, v_{[-i]}) v_i, \tag{55}$$

using the notation $v_{[-t]} := (v_1, \ldots, v_{t-1}, v_{t+1}, \ldots, v_n)$. We now show that all $\tilde{f}_i$'s can be chosen to the same functions. In order to do so average over permutations again:

$$h(v_1, \ldots, v_n) = \frac{1}{n!} \sum_{i=1}^{n} \sum_{\sigma \in S_n} \tilde{f}_i(v_{\sigma(i)}, v_{[-\sigma(i)]}) v_{\sigma(i)}, \tag{56}$$

$$= \frac{1}{n!} \sum_{i=1}^{n} \left( \sum_{j=1}^{n} \sum_{\sigma \in \Pi_{i \to j}} \tilde{f}_i(v_j, v_{[-j]}) \right) v_j, \tag{57}$$

$$= \frac{1}{n!} \sum_{j=1}^{n} \left( \sum_{i=1}^{n} (n-1)! \, \tilde{f}_i(v_j, v_{[-j]}) \right) v_j, \tag{58}$$

$$= \sum_{j=1}^{n} \left( \frac{1}{n} \sum_{i=1}^{n} \tilde{f}_i(v_j, v_{[-j]}) \right) v_j, \tag{59}$$

Therefore we define $\hat{f}$ such as

$$\hat{f}(w, w_1, \ldots, w_{n-1}) = \frac{1}{n} \sum_{i=1}^{n} \tilde{f}_i(w, w_1, \ldots, w_{n-1}), \tag{60}$$

a $O(d)$-invariant function that is permutation invariant with respect to the last $n-1$ inputs. The computation in (59) shows that:

$$h(v_1, \ldots, v_n) = \sum_{j=1}^{n} \hat{f}(v_j, v_{[-j]}) v_j, \tag{61}$$

which proves the theorem. $\qquad \square$

**Proof of Proposition 11:** Let $\sigma \in S_n$ and $(h_1, \ldots, h_n) = h : (\mathbb{R}^d)^n \to (\mathbb{R}^d)^n$ a permutation-equivariant function. Then

$$h(\sigma \star (v_1, \ldots, v_n)) = \sigma \star h(v_1, \ldots, v_n) \tag{62}$$

$$h(v_{\sigma(1)}, \ldots, v_{\sigma(n)}) = (h_{\sigma(1)}(v_1, \ldots, v_n), \ldots, h_{\sigma(n)}(v_1, \ldots, v_n)) \tag{63}$$

$$(h_1(v_{\sigma(1)}, \ldots, v_{\sigma(n)}), \ldots, h_n(v_{\sigma(1)}, \ldots, v_{\sigma(n)})) = (h_{\sigma(1)}(v_1, \ldots, v_n), \ldots, h_{\sigma(n)}(v_1, \ldots, v_n)) \tag{64}$$

Since $h_i : (\mathbb{R}^d)^n \to \mathbb{R}^d$ are O($d$)-equivariant (or continuous O($1, d-1$)-equivariant) we have that for all $i$ there exists $f_t^i : (\mathbb{R}^d)^n \to \mathbb{R}$ O($d$)-invariant (or O($1, d-1$)-invariant) such that:

$$h_i(v_1, \ldots, v_n) = \sum_{t=1}^{n} f_t^{(i)}(v_1, \ldots, v_n) v_t. \tag{65}$$

Combining (64) and (65) we get

$$h_i(v_\sigma(1), \ldots, v_\sigma(n)) = \sum_{t=1}^{n} f_t^{(i)}(v_{\sigma(1)}, \ldots, v_{\sigma(n)}) v_{\sigma(t)} \tag{66}$$

$$h_{\sigma(i)}(v_1, \ldots, v_n) = \sum_{t=1}^{n} f_t^{(\sigma(i))}(v_1, \ldots, v_n) v_t. \tag{67}$$

Taking $t = \sigma^{-1}(j)$ in (66) we get $t = j$ in (67) and matching coefficients we get

$$f_{\sigma^{-1}(j)}^{(i)}(v_{\sigma(1)}, \ldots, v_{\sigma(n)}) = f_j^{(\sigma(i))}(v_1, \ldots, , v_n). \tag{68}$$

$\qquad \square$

## F  Einstein summation notation

We can interpret the results from Section 3 as coming from the symmetries encoded in the Einstein summation rules.

An important early realization in differential geometry and in general relativity was that an enormous class of generally covariant forms can be written in a form that is commonly known (incorrectly perhaps) as "Einstein summation notation" [25]; it is a subset of the prior Ricci calculus [76]. This notation is a method for finding and checking equivariant quantities useful in physical laws: We imagine that we have O($d$)-equivariant vectors $u$, $v$, $w$, and each has $d$ components such that $[u]_i$ is the $i$th component of $u$. If we write products with repeated indices like $[u]_i [v]_i$, the sum rule is that the repeated index $i$ is summed, so this corresponds to a O($d$)-invariant scalar product (dot product or inner product) of the vectors $u$ and $v$.

$$[u]_i [v]_i := \sum_{i=1}^{d} [u]_i [v]_i = u^\top v \,, \tag{69}$$

where the first := in (69) defines the summation notation and the second = relates this sum to the linear-algebra operation on two column-oriented vectors ($d \times 1$ matrices). All indices appear either once (unsummed) or twice (summed) but never more than twice:

$$\underbrace{[u]_i [v]_i [w]_i}_{\text{indices can only appear once or twice!}} = \text{undefined} \,. \tag{70}$$

If there is one unsummed index in an expression, as in $[u]_i [v]_i [w]_j$, then the result will be an O($d$)-equivariant vector:

$$[u]_i [v]_i [w]_j \equiv \left( \sum_{i=1}^{d} [u]_i [v]_i \right) w_j \equiv (u^\top v)\, w \,. \tag{71}$$

If there are $\ell$ unsummed indices, then the expression is an O($d$)-equivariant order-$\ell$ tensor.

This notation also reveals that the directionality of the output is entirely encoded in the input vectors and their combination. This is counter-intuitive given some physics expressions, such as the electromagnetic force law explained in (15); it contains a cross product that typically requires a particular coordinate system to determine the direction of the result. In this case, the direction of the force cannot depend on the coordinate system; the second cross product saves us, but it is not immediately clear how. With Einstein notation, we can express the vector triple product in a coordinate-free way. We first rewrite the $d = 3$ cross product (pseudo-vector product) $a \times b$ in terms of the maximally anti-symmetric rank-3 tensor in $d = 3$ (the Levi-Civita symbol) $\epsilon_{ijk}$:

$$f(u, v, w) = (u \times v) \times w \tag{72}$$
$$f_n = [u]_j [v]_k [w]_m\, \epsilon_{ijk}\, \epsilon_{imn} \,. \tag{73}$$

The properties of the anti-symmetric tensor products are such that this product can be re-written as

$$f_n = [u]_j [v]_k [w]_m\, [\delta_{jm}\, \delta_{kn} - \delta_{jn}\, \delta_{km}] \tag{74}$$
$$f_n = [u]_i [v]_j [w]_i - [u]_j [v]_i [w]_i \tag{75}$$
$$f(u, v, w) = (u^\top w)\, v - (v^\top w)\, u \,, \tag{76}$$

where $\delta_{ij}$ is the Kronecker delta. We thus see that the cross product can be expressed entirely in terms of scalar products ($u^\top w$ and $v^\top w$) and the input vectors ($v$ and $u$), and that this can only be done by employing the anti-symmetric tensor.

Einstein notation is often credited with making physical-law expressions more compact or brief. But what's important about the notation is that if the rules are obeyed, the notation can produce only equivariant objects in the theory.

Under these summation rules, all O($d$)-invariant scalar expressions that can be made from polynomial expressions of vectors will include only terms that use even numbers of vectors. All these terms can be rearranged to be written as products of invariant scalar products. That is, any scalar function that can be written as a polynomial of vectors (or a function of such polynomials) can be written

in terms only of available scalar products. This result corresponds very directly to Lemma 1. For concreteness, here is an example 4-vector scalar form, written in summation notation, reordered as an inner product of tensors or as a simple product of scalars:

$$\underbrace{[u]_i\,[v]_j\,[w]_i\,[z]_j}_{\text{4-vector scalar polynomial term}} = \underbrace{([u]_i\,[v]_j)\,([w]_i\,[z]_j)}_{\text{inner product of order-2 tensors}} = \underbrace{([u]_i\,[w]_i)\,([v]_j\,[z]_j)}_{\text{product of scalars}}\,. \tag{77}$$

On the other hand, all O($d$)-equivariant vector expressions that can be made from polynomial expressions of vectors will include only terms that use odd numbers of vectors. These terms can be rearranged by the vector with the unsummed index. Once they are rearranged this way, the expression becomes the input vectors times polynomial expressions of scalar products. This demonstrates that any O($d$)-equivariant vector expression constructible in the notation will lie in the subspace spanned by the input vectors. It also demonstrates that the O($d$)-invariant scalar coefficients multiplying those vectors must themselves be constructible from polynomials (or functions of polynomials, it turns out) of scalar products. What is also true but to the best of our knowledge doesn't appear explicitly stated in the physics literature is that every O($d$)-equivariant polynomial vector function can be expressed in Einstein notation. This result correspond very directly to Proposition 4.

When the metric is non-trivial, we must face the covariant/contravariant distinction, where vector components might be pre-multiplied by the metric or not. In this context, there are components $[u]_i$ of the covariant vectors and components $[u]^i$ of the contravariant vectors; the Einstein summation notation rules obtain the additional rule that repeated indices must belong to covariant-contravariant pairs:

$$[u]_i\,[v]^i \equiv \sum_{i=1}^{d}[u]_i\,[v]^i \equiv \sum_{i=1}^{d}\sum_{j=1}^{d}[u]_i\,[\Lambda]^{ij}\,[v]_j \tag{78}$$

$$[v]^i \equiv \sum_{j=1}^{d}[\Lambda]^{ij}\,[v]_j\,, \tag{79}$$

where the $[\Lambda]^{ij}$ are the components of a $d \times d$ Hermitian metric tensor. In the case of the Lorentz group, $d = 4$ and the metric is diagonal, with elements $(-1, 1, 1, 1)$ on the diagonal. In the case of the curved spacetime of general relativity, $d = 4$ and the metric is (in general) a function of spatial position and time.

We expect that these results will have generalizations for scalar, vector, and tensor functions of scalar, vector, and tensor inputs.

## G   Connections with low-rank matrix completion

Given a rank $d$, $n \times n$ matrix $M$, Example 4 in [71] shows that $M$ is almost always uniquely determined by the entries $\Omega(M) := M_{i,i+s}\ i = 1,\dots,n,\ s = 0,\dots,d$, considering the indices "wrap around" (i.e., $M_{i,n+s}$ corresponds to $M_{i,s} = M_{i,n+s(\mathrm{mod}\ n)}$). In particular invariant functions $f : (\mathbb{R}^d)^n \to \mathbb{R}^\ell$, can be expressed as (5)

$$f(v_1,\dots,v_n) = g(M) = \tilde{g}(\langle v_i, v_{i+s(\mathrm{mod}\ n)}\rangle_{1 \le i \le n,\ 0 \le s \le d}) = \tilde{g}(\Omega(M)), \tag{80}$$

where $M$ is either $V^\top V$ or $V^\top \Lambda V$. Therefore, any function $\tilde{g} : \Omega(M) \to \mathbb{R}^\ell$ uniquely determines $f$ on almost all possible inputs $v_1,\dots,v_n$. This observation provides a parameterization for Lorentz and orthogonal-invariant and equivariant functions with a small number of scalars.

We note that the results from [71] don't assume the matrix $M$ is positive semi-definite, only that it is low rank. This is useful since $M$ is not positive semi-definite in the Minkowski case. We also remark that the learning of the functions (3) and (5) can be done from $\Omega(M)$ without ever needing to compute all the scalars.

One disadvantage of the subset of scalars proposed in this section is that the sampling procedure $\Omega(M)$ is not permutation invariant. One of our future goals is to find a set of permutation invariant scalars that are universally expressive.

## H  Model-building considerations

The results in Section 3 provide a simple characterization of all scalar functions and vector functions that satisfy important symmetries (see Table 1) for classical physics and special relativity. Here we discuss how we might use this to design and build a model.

First we need to identify the inputs, what groups are acting, and how the group action affects the inputs and outputs. For example, the translation group acts differently on position vectors than displacement vectors (see Table 2) than other kind of vectors (velocities, accelerations, fields, etc). Moreover, permutations don't typically act on the entire set of inputs but on tuples of inputs like the individual-particle charge, position, and velocity $(q_i, r_i, v_i)$ in the electromagnetic example in Section 4.

The causal and group-action structure of the model must correspond to the structure of the physics problem. For example, suppose we are interested in solving an $n$-body problem in which we predict the trajectories of a set of interacting charged particles. The target functions $h_i()$, which predict the position of particle $i$ at time $t$, can be described as functions of that particle's charge, position, and velocity, and the set of charges, positions, and velocities of all the others. The group actions (rotation, translation, and permutation) can be written as:

$$h_i(q_i, Q\,r_i, Q\,v_i, (q_j, Q\,r_j, Q\,v_j)_{j=1; j\neq i}^n) = Q\,h_i(q_i, r_i, v_i, (q_j, r_j, v_j)_{j=1; j\neq i}^n) \tag{81}$$

$$h_i(q_i, r_i + w, v_i, (q_j, r_j + w, v_j)_{j=1; j\neq i}^n) = h_i(q_i, r_i, v_i, (q_j, r_j, v_j)_{j=1; j\neq i}^n) + w \tag{82}$$

$$h_i(q_i, r_i, v_i, (q_{\sigma_i(j)}, r_{\sigma_i(j)}, v_{\sigma_i(j)})_{j=1; \sigma_i(j)\neq i}^n) = h_i(q_i, r_i, v_i, (q_j, r_j, v_j)_{j=1; j\neq i}^n) \tag{83}$$

$$h_{\sigma(i)}(q_i, r_i, v_i, (q_j, r_j, v_j)_{j=1; j\neq i}^n) = h_i(q_i, r_i, v_i, (q_j, r_j, v_j)_{j=1; j\neq i}^n)\,, \tag{84}$$

where $\sigma$ is any permutation on $n$ elements, and $\sigma_i$ is a permutation that fixes $i$, i.e., $\sigma_i(i) = i$. The primary results of this paper imply that, in this $n$-body case, the rotation symmetries can be enforced by constructing the invariant scalar products, and building functions thereof. The translation symmetries can be enforced by taking positional differences with respect to $r_i$ prior to constructing the scalars. The first permutation symmetry (83) can be enforced by working on a set-based neural network (or graph neural network), and the second permutation symmetry (84) can be enforced by taking all the functions $h_i = h$ to be identical. In general, graph-based or set-based methods are probably good frameworks for physics problems [19, 79], and the dynamics can probably be implemented with a form of message-passing.