# OpenReview forum: "Scalars are universal: Equivariant machine learning, structured like classical physics"
_NeurIPS.cc/2021/Conference — NeurIPS 2021 Poster_

### Official Review · Reviewer_K6Cj · 2021-07-16

**Rating:** 6
**Confidence:** 2

**Summary:**

The paper introduces characterizations of invariant and equivariant functions with respect to various symmetries. Starting from basic invariance with respect to rotation, translation, and permutation, the authors build up their toolkit to parametrize equivariant functions with respect to common symmetry groups in physics, including Euclidean, Lorentz, and Poincaré groups. The authors finally provide an example of applying their ideas to build a message passing neural network for learning the symmetries in the n-body problem.

**Limitations And Societal Impact:**

The paper contains a limitation section that addressed many questions that I have. My remaining concerns are listed in the "Main Review" section.

**Main Review:**

**Quality**: I checked most of the proofs and to the best of my knowledge, they seem to be correct.

**Clarity**: The paper is generally well-written in terms of motivation and related work. The section on equivariant maps can be made easier to read if the Definition, Lemma, and Proposition titles are boldfaced, to distinguish these from the main text. Many of the sum symbols in the Appendix are missing upper and lower indices, and these should be added.

**Originality**: The authors collect the results on the characterization of equivariant functions in a different perspective from most previous papers, which focus exclusively on a few specific physics problems and their corresponding functions. I am not entirely familiar with the literature to comment on the originality of the lemmas, though at least some results in the collection seem to be well-known or easy extensions (e.g. that of rotation and translation), while others seem to be original.

**Significance**: On the good side, the paper provides a good collection of results for the characterization of invariant and equivariant functions with respect to various symmetries that may be useful for future research in this area. The message passing neural networks framework also gives a good way of incorporating different types of symmetries in one network.

My main concerns/suggestions for improvement are as follows:

(1) The section on estimating the number of scalars only gives a high-level discussion, which is not very satisfying. This can be made better if the authors can extend their discussion into several theorems, and give an explicit comparison to results for previous networks (which are mentioned in Appendix A, though also given in very high level without explicit theorems).

(2) While the framework makes sense to me in the theoretical sense, I am rather unconvinced on the practicality of the method. The paper would be much more solid if the authors can provide an empirical comparison of their proposed model with the current state-of-the-art models.

**Additional Comments/Questions**: Is there proof for Proposition 9? I do not seem to see the proof either in the paper or the appendix.


**Time Spent Reviewing:**

10

---

> ### Author Response · Authors · 2021-08-09
> **Response to reviewer**
>
> Thank you for your comments. We agree the connection with matrix completion is a very interesting research direction where mathematical theory can be developed and theorems can be proved. This is actually part of a research program we are starting.
>
> We agree with you and most of the reviewers that the paper would be greatly improved if implementation and experimental results were provided. For this reason we provide an implementation of the model and experimental results in an anonymized github repository: https://github.com/Pamplemousse-Elaina/Comparison_EMLP We find that our implementation performs better than its competitors. We will modify the manuscript to include these numerical experiments, their results, and discussion.
>
> Proposition 9. We didn't spell out the proof, but the characterization of Poincaré symmetries can be proven in the same way as the characterization of Euclidean symmetries where the standard inner product must be replaced by the Minkowski inner product. We will update the paper to explain that point.

---

### Official Review · Reviewer_D3qF · 2021-07-16

**Rating:** 6
**Confidence:** 4

**Summary:**

The paper presents a mathematical idea for how to define a universal class of invariant and equivariant functions. These functions act on a collection of d-dimensional points, and output a single number or a vector. The group acts diagonally, applying the same transformation to each point. Using results from classical invariant theory, it is shown that such in/equivariant functions can always be written as a function of the collection of inner products between the d-dimensional vectors (or some subset thereof). This general idea is worked out for the groups that are most important in physics problems.

**Limitations And Societal Impact:**

Yes

**Main Review:**

I found this paper very interesting to read. It contains a thorough mathematical study of in/equivariant functions of collections of points, and suggests a research direction that is different from the current mainstream approach in geometric/equivariant deep learning. The paper is well written and should be accessible to most machine learning researchers with a bit of effort.

Although clearly a lot of work has gone into this work and the paper feels polished enough, the lack of implementation and experiments makes me question whether the paper is ready for prime time. In principle I am happy to accept purely theoretical papers, but in this case the main contribution is an *idea* for how to build models for learning problems in physics, that may or may not work well in practice. Although it is nice to know that this class of models/functions is universal in principle, the more important question is how well it works in practice. We cannot know this without experiments, and there are some reasons to think that this approach may not work that well (see below).

At its core, this paper suggests going back to the “feature engineering followed by learning” approach that was popular before deep learning where “end-to-end representation learning” is used. The features in this case are invariant inner products, which are shown to capture all the relevant information. Although feature engineering might work for the type of problems of interest in this work, we can’t know without experiments, and the overall trend of the field suggests that the DL approach is often superior.

One potential weakness of the proposed approach is that although it leverages the geometric prior of symmetry, it ignores scale separation / locality (See the book “Geometric Deep Learning: Grids, Groups, Graphs, Geodesics & Gauges” which proposes these as the two key geometric priors of geometric DL). That is, all points interact with all other points immediately at the input layer / feature computation stage. For many problems it may make more sense to have nearby points interact first, apply coarsening / pooling, and then gradually allow longer range interactions, as happens for instance in a convolutional network.

The paper suggests that one could make the whole model into a polynomial. Although many classical physics situations can be captured in this way,polynomials can be unstable to deformations that are not part of the group, and it is not clear if such models will share the benign optimization landscapes that we find with neural networks built as compositions of learnable linear layers and pointwise (non-polynomial and often non-analytic) nonlinearities.

Of course these are just speculations and I would encourage the authors to test their idea on real-world problems to settle this question.

I would note that the idea presented in this paper can be generalized via the use of computational invariant theory. The paper assumes a particular kind of action of the group (v1, …, vn) -> (g v1, …, g vn). If we stack the vi into a long vector, we see that we have a representation of the group that is block diagonal with identical blocks. Instead one could allow for an arbitrary representation. In case the group is algebraic (specified by a polynomial constraint such as Q^T Q = I) and the representation is algebraic, there exist algorithms that will automatically generate, via the use of Grobner bases, a (minimal) set of generators for the ring of polynomial invariants.

A final quibble, which does not affect my overall judgement of the paper, pertains to the use of the word “gauge”. Gauge symmetry should not be confused with “coordinate freedom” as done in the abstract, or with global symmetries such as those in table 1 & 2. Properly understood, a gauge transformation is an automorphism of a principal bundle that fixes the base space (see e.g. “Mathematical Gauge Theory” by Hamilton). The general setting is one where we have a base space (e.g. space time), with at each point a “fiber” (a vector space, whose dimensions may or may not be related to directions in the base space). A gauge transformation applies a change of frame to each fiber independently, and (for continuous base spaces) the group of gauge transformations is thus infinite dimensional, unlike the groups considered here. The groups considered in this paper act on the base space R^d, and thus do not fix the base space as a gauge transformation would.

In conclusion, although this is a very substantial and interesting piece of work, it is still in the idea stage. Perhaps one can argue that the ideas are interesting enough to warrant publication as is, but personally I think that it would be better to first validate the ideas before publishing at a conference.

Typos
p3 : “deep nets” -> “deep sets”


Post rebuttal edit:
Since the authors have now provided an implementation that appears to work well, I have increased my score.


**Time Spent Reviewing:**

3

---

> ### Author Response · Authors · 2021-08-09
> **Response to reviewer**
>
> We appreciate this thoughtful and thorough review. We agree that the biggest limitation with the paper as submitted is the lack of implementation and experiments. For this reason we provide an implementation of the model and experimental results in an anonymized github repository: https://github.com/Pamplemousse-Elaina/Comparison_EMLP We find that our implementation performs well. We will modify the manuscript to include these numerical experiments, their results, and discussion.
>
> On the “feature engineering” point: We did not intend to recommend feature engineering followed by learning! We have modified the paper to remove any such suggestion. In particular, the selection of a subset of available scalars is not intended to be a feature selection, but instead it is a method to deliver *all possible features* without actually delivering all scalars.
>
> The question of whether this model respects true local gauge symmetries or just global coordinate symmetries depends on the scope of the input vectors. Indeed, the Einstein summation notation was invented for problems in general relativity, where the gauge symmetry is true and local. In physics, if we have local gauge symmetries then the functions that appear are functions of only local field values and our characterization still works. If two input vectors are spatially separated and are inputs to a locally invariant function then one vector should need to be propagated to the location of the other vector before the function is evaluated. The rules of vector propagation would also need to obey our characterization. This is an important point and we thank the reviewer for pointing it out. We will add a discussion in the revised manuscript related to this point.

---

> > ### Comment · Reviewer_D3qF · 2021-08-16
> > **Experiments**
> >
> > Dear authors,
> >
> > Thanks for providing an implementation and experiments. Although I have not thoroughly studied the code, I find the experiments to be quite convincing. The proposed method outperforms a recent method by Finzi et al. in several of the datasets considered in that paper. The code also seems quite simple. Only it seems like the O(1,3) / Lorentz experiments are missing. It would be nice to add these, even if the proposed method does not outperform the one proposed by Finzi et al.
> >
> > Regarding feature engineering: I still think that this characterization is appropriate, for the reasons mentioned in my review. The invariant inner products are features that are designed by hand to have desirable properties (namely invariance and completeness). Feature computation is fixed, and followed by learning. I also still believe that the fact that these features fuse all information (including interactions between far away points) at once could be a downside for some problems. It would be interesting to find such problems, and also to discuss here or in the paper why early fusion / lack of a locality prior is not an issue for the datasets considered. Despite the fact that this work is essentially about feature engineering, the results speak for themselves and I don't think we should reject a paper only because it goes against the mainstream philosophy of the day. Hence I have updated my score.
> >
> > I did not really follow your answer regarding gauge symmetries but I still believe the current method should not be viewed as a gauge equivariant method, and would suggest to consider changing "gauge" to "group". One simple way to see this is that the groups considered here are finite (low) dimensional groups, whereas the group of gauge transformations is typically infinite dimensional (when applied to fields on a space with an infinite number of points like a manifold) or at least should allow for a freely varying transformation at each point of the space. Also, the concept of a local vs global gauge symmetry is a confusion in the physics literature: whether a particular gauge transformation is global or not depends on the coordinate system used.

---

> > > ### Author Response · Authors · 2021-08-17
> > > **Response on experiments**
> > >
> > > Thank you. We are currently implementing the Lorentz experiments, we expect to update the Github tonight.
> > >
> > > Regarding feature engineering, we see your point. We didn't think of it as feature engineering because it is a mathematical characterization of the invariant functions, but we understand it can be interpreted as such.
> > >
> > > Finally, we agree we should change the title to remove the word gauge. We posted a general comment about it.

---

### Official Review · Reviewer_9FH3 · 2021-07-16

**Rating:** 7
**Confidence:** 2

**Summary:**

The authors theoretically discuss the universal approximating function space for various symmetries that frequently hold in classical physics. They find that all invariant scalar functions and equivariant vector functions can be expressed by a collection of scalar products and contractions. The authors’ findings can provide a simple characterization of such invariant and equivariant functions. It would be particularly useful when modeling and learning physical phenomena that naturally manifest such symmetries.

**Limitations And Societal Impact:**

The authors discuss the limitation of their work in Section 6 of the paper.

**Main Review:**

Originality and quality: The authors clearly provide two important facts on representing invariant and equivariant functions with “scalars”. Then, they apply these facts for truly revealing some important symmetries, e.g., orthogonal, rotational, translational, Euclidean, Lorentzian, Poincare, and permutations. To me, these findings are interesting and technically sound.

Clarity: This paper is generally well-written. The motivation and main contribution of the paper are clearly presented. Although I am a beginner at invariant theory, I was able to follow the paper to a certain degree.

Significance: Enforcing symmetries is highly desirable for learning the laws of physics. This paper contributes a solid theoretical background for such a job. The authors also discuss the practical usefulness of their findings in Section 5 (in terms of scalability), Section 6 (in terms of limitations), and Section 7 (in terms of practical model building considerations), which makes their paper be highly appealing for practitioners like me.

Overall, I think this paper is a good contribution to both machine learning and physics communities. Therefore, I would like to vote to accept this paper. However, since I am not an expert in this field, I cannot exhaustively evaluate this contribution, especially in terms of originality and quality. Thus, my rating may be updated after discussing with the authors or other expert reviewers.


**Time Spent Reviewing:**

6

---

> ### Author Response · Authors · 2021-08-09
> **Response to reviewer**
>
> Thank you for your kind review.

---

### Official Review · Reviewer_Rogp · 2021-07-17

**Rating:** 6
**Confidence:** 4

**Summary:**

This paper makes an interesting observation using standard ideas from classical invariant theory. Specifically, that the design of networks equivariant to various symmetry groups in classical physics can be done using a collection of invariant scalars. This suggests an attractive alternative approach for the design of equivariant networks. This approach circumvents the need for careful usage of the irreps, solving particular Clebsch-Gordan decompositions (which might not also be available) and so on; something inherent in a lot of current work in the area. The characterization of equivariant functions using such scalars are provided for various groups relevant to classical physics i.e. the orthogonal, rotation, translation, Euclidean, Lorentz, Poincare and permutation groups. Two examples relevant to physics are provided along with some observations on model building considerations.

**Limitations And Societal Impact:**

Limitations are discussed in detail and with candor.

**Main Review:**

Much recent effort has gone into the careful design of group/gauge equivariant neural networks that incorporate various task specific symmetries. A lot of activity has also been seen in the use of such networks in various applications in physics. Some of these approaches require irreducible representations of the group under consideration and also need to solve the relevant Clebsch-Gordan problem, which might not always be possible. The main contribution of the paper is the observation that for an array of groups relevant to classical physics (orthogonal, rotation, translation, Euclidean, Lorentz, Poincare and permutation), such approaches are not necessary. It provides a tractable characterization for the space of equivariant functions (to these groups) by showing that they can be constructed only from a subset of invariant scalars (scalar products and contractions of the input vector or tensor).

The key to the above observation are some classical and very well known results from invariant theory (going back to, and stated in their earliest forms, by Weyl). For n input vectors v_1,...v_n in R^d, Lemma 1 (which is due to Weyl) shows that that O(d) invariant functions are functions of the invariant scalars v_i ^T v_j. Lemma 2 gives an analogous result for SO(d) which uses d x d determinants of the inputs when arranged in a d by n matrix V. Proposition 4 extends the result in Lemma 1 to equivariant functions, while proposition 5 does so for Lemma 2. Similar results are shown for E(d), Lorentz, Poincare and permutation groups. Two illustrative examples are provided (total mechanical energy and electromagnetic force law), showing how to formulate them in terms of such invariant scalars -- which can then used in a machine learning system. Section 5 addresses the question on how many scalars might be necessary, leveraging results from literature adjacent to matrix completion and compressed sensing. This section shows that it is not necessary to use all the scalars obtained from pairwise operations, but only a relevant subset of them. Finally some model building considerations are discussed.


Central to the paper is essentially the observation that the design of equivariant networks in the context of physics (and more broadly) can be greatly simplified using the above ideas. Another relevant contribution is its scope -- it focuses on groups important in classical physics. The paper is also extremely well written, with most of the ideas fleshed out in admirable detail in the appendix. However, despite these, I feel conflicted about recommending acceptance. The results in the paper are essentially well known or minor extensions of classical results in invariant theory -- which by itself is not a problem. As I have mentioned above, the chief import of the paper is the observation that these can be relevant to constructing equivariant NNs without resorting to complex acrobatics with regard to the irreps etc. The use of such invariant scalars in the context of physics and machine learning does appear from time to time (although granted not for these set of groups and the broader scope that the paper covers). For instance see: 1. Lorentz- and permutation-invariants of particles, Ben Gripaios, Ward Haddadin, and Christopher G Lester, Journal of Physics A: Mathematical and Theoretical, Volume 54, Number 15; 2. Invariant polynomials and machine learning, Ward Haddadin, arXiv:2104.12733v1. Papers such as "Moment Tensor Potentials" by Shapeev are also relevant.

Some of these considerations make numerical experiments comparing the framework, on at least one standard task, quite pressing and essential (even if it has enough material in it for others to build systems using these ideas and actually leading to publications themselves). Some experimental validation is essential because it is not at all clear whether using scalars in the way described will lead to practical networks. The framework described in the paper will also need reformulation of classical theories at times. It is also not clear whether such reformulations will lead to numerically stable networks. I am worried that the proverbial devil will be in the (implementation) details, and that's where a lot of work will be required. Because of these considerations, despite liking the paper, I am currently leaning towards a score of 5. If during the rebuttal, the authors can convince me otherwise, I would be more than happy to raise my score.

Minor comments:

- "convolutional symmetry" is a phrase that appears in the paper in a few places. Is this standard terminology? Unless I am missing something, this doesn't sound quite right.
- line 100: typo -- looks like it should say deep sets (instead of deep nets).
- lines 119 - 122: This might be a matter of opinion. But I don't quite see how the universal approximation theorem using non-polynomial activations ([46] in the paper) inspired some of the recent equivariant models. In the earlier equivariant NN papers (before universality results for equivariant networks started to appear), these results are either not cited, or had perfunctory citations.
- equation 17 seems to have a typo. More specifically, it overcounts the energy. The second summation should go over j > i

**Time Spent Reviewing:**

4 hours

---

> ### Author Response · Authors · 2021-08-09
> **Response to reviewer**
>
> This is a very nice referee report and we really appreciate it.
>
> We agree with the main point made by the reviewer: the devil is in the implementation details. For this reason we provide an implementation of the model and experimental results in an anonymized github repository: https://github.com/Pamplemousse-Elaina/Comparison_EMLP We find that our implementation performs well and it is numerically stable, although of course our implementation has only been tested on a few toy problems (we reproduced the experiments 1,2 and 4 from Finzi et al. ICML 2021). We have modified the manuscript to include these numerical experiments, their results, and discussion.
>
> We thank the referee for noting additional literature relevant to our work, and will cite these accordingly. In particular, the Gripaios (2021) paper and the very recent work Haddadin (2021) are very relevant to our formulation. We plan to discuss them in our revision of this paper.
>
> The referee’s minor comments (thank you!) will be addressed and fixed in the revised version. In particular we have corrected the terminology around convolutional symmetry, which refers to the symmetries implemented by convolutional neural networks.

---

> > ### Comment · Reviewer_Rogp · 2021-08-25
> > **Experiments**
> >
> > I have looked at the code and experiments presented, and although they are quite preliminary from my point of view. I think they are fairly convincing, performing better than the method of Finzi et al. While I would like to see more validation, I have raised my score by one notch.

---

### Official Review · Reviewer_WMkN · 2021-08-01

**Rating:** 4
**Confidence:** 3

**Summary:**

The paper presents an approach to parameterization of functions that are invariant or/(and) equivariant under the action symmetries commonly present in physical systems. The construction of such parametrization is based on known mathematical results relating the dependence structure of invariant/equivariant maps with scalar and vector outputs.

Authors explicitly construct parameterizations for cases of scalar functions that are: (1) invariant and equivariant under O(d) and SO(d) groups; translation invariant; Lorentz symmetry; as well as several combinations. Relevance to classical laws of nature is provided via examples of physical laws in classical mechanics and electromagnetism.

Authors discuss limitations and caveats of the presented approach and note that the original construction can be improved in terms of the number of input features by relying on the result from the low-rank matrix completion problem.

**Limitations And Societal Impact:**

This work does not have negative societal impact. The limitations are addressed adequately.

**Main Review:**

The paper discusses a relevant topic of efficient encoding of symmetry properties into neural networks by construction. The paper is written clearly and I especially would like to highlight a detailed section on limitations of the proposed method.

While the presented approach is perfectly valid, I am not convinced that it is novel and significant enough to be of interest to a wider community. The idea of building-in invariances and equivariances by construction has been present in research papers on using ML for natural sciences for some time (e.g. [1]). While there is also interest in exploiting the symmetry properties in many other areas of machine learning besides science oriented applications, the presented approach is rather challenging to generalize to arbitrary/apriori unknown symmetries.

[1] Ling, J., Kurzawski, A., & Templeton, J. (2016). Reynolds averaged turbulence modelling using deep neural networks with embedded invariance. Journal of Fluid Mechanics, 807, 155-166.


**Time Spent Reviewing:**

4

---

> ### Author Response · Authors · 2021-08-09
> **Response to reviewer**
>
> Thank you for pointing out the reference Ling et al. (2016). We agree that Ling et al. presents a method similar to ours: it embeds Galilean equivariance in a neural network, using a basis of invariant tensors. However, our work is more general, extending to many other symmetry groups; we also focus on vector inputs, while this paper uses tensor inputs, giving different and novel results. We will add a discussion of this to the revised manuscript.
>
> Regarding the other comments we'd like to point out that while there is a literature in building invariant machine-learning methods, the method we present is novel. And we agree we cannot address arbitrary symmetries, nor can any other method in this area.

---

### Author Response · Authors · 2021-08-17
**Title change**

After careful consideration we agree with the reviewers that our model doesn't cover all gauge symmetries. We believe our model could be made general enough if we replace the local metric by any position-dependent metric $\Lambda_x$. However, we agree that we didn't write a full gauge-invariant/equivariant model so we are removing the word gauge from the title and changing the manuscript accordingly.

The title of the article will be:
Scalars are universal: Equivariant machine learning, structured like classical physics

We plan on working on a full gauge-equivariant model in future work. Thank you very much for your feedback.

---

### Decision · Program_Chairs · 2021-09-28

**Decision:**

Accept (Poster)

**Comment:**

Equivariant neural networks are very important for applications of ML to physics problems. This paper explores the idea of building equivariant neural networks from invariant scalar functions which are simpler to build. This principle is well-known and commonly used in physics but has not been extensively explored in the ML literature for equivariant model building.

The paper was originally presented as a general mechanism for building Gauge equivariance --- however all the presented theory only considers global (coordinate) symmetries. A few extra ingredients are needed to build Gauge equivariance on top of the presented theory such as the parallel transport of tensors at different locations to build local invariants. The authors promised the paper was modified to its correct scope.

Concerns have been raised regarding the lack of experiments in the original submission, but the authors added an implementation of these ideas and experiments to an anonymised git repository. The experimental results are a bit preliminary but look promising.

**Consistency Experiment:**

NeurIPS has a long history of experimentation. In 2014, NeurIPS ran an experiment in which 10% of submissions were reviewed by two independent committees to quantify the randomness in the review process. This year, we repeated a variant of this experiment to see how the quality of the review process has changed over time.  This paper was part of the experiment and was therefore assigned to two committees (consisting of reviewers, an Area Chair, and a Senior Area Chair) that reached independent decisions.  If both committees made the same recommendation, this recommendation was followed. If a single committee recommended acceptance, the paper was accepted (with the exception of a few cases in which the other committee identified what we considered a fatal flaw, e.g., an error in a key result).

Both committees reached the same decision: **Accept (Poster)**

The other committee assigned to the paper recommended **Accept (Poster)**.  You can find the other set of reviews, along with any follow up discussion with the authors here:
https://openreview.net/forum?id=NqYtJMX9g2t